# A cell wall synthase accelerates plasma membrane partitioning in mycobacteria

Takehiro Kado[1], Zarina Akbary[2], Daisuke Motooka[3], Ian L Sparks[1],
Emily S Melzer[1], Shota Nakamura[3], Enrique R Rojas[2], Yasu S Morita[1,4]*[†],
M Sloan Siegrist[1,4]*[†]

[1]Department of Microbiology, University of Massachusetts Amherst, Amherst, United States; [2]Department of Biology, New York University, New York, United States; [3]Department of Infection Metagenomics, Research Institute for Microbial Diseases, Osaka University, Osaka, Japan; [4]Molecular and Cellular Graduate Program, University of Massachusetts Amherst, Amherst, United States

**Abstract** Lateral partitioning of proteins and lipids shapes membrane function. In model membranes, partitioning can be influenced both by bilayer-intrinsic factors like molecular composition and by bilayer-extrinsic factors such as interactions with other membranes and solid supports. While cellular membranes can departition in response to bilayer-intrinsic or -extrinsic disruptions, the mechanisms by which they partition de novo are largely unknown. The plasma membrane of *Mycobacterium smegmatis* spatially and biochemically departitions in response to the fluidizing agent benzyl alcohol, then repartitions upon fluidizer washout. By screening for mutants that are sensitive to benzyl alcohol, we show that the bifunctional cell wall synthase PonA2 promotes membrane partitioning and cell growth during recovery from benzyl alcohol exposure. PonA2's role in membrane repartitioning and regrowth depends solely on its conserved transglycosylase domain. Active cell wall polymerization promotes de novo membrane partitioning and the completed cell wall polymer helps to maintain membrane partitioning. Our work highlights the complexity of membrane–cell wall interactions and establishes a facile model system for departitioning and repartitioning cellular membranes.

**\*For correspondence:**
ymorita@umass.edu (YSM);
siegrist@umass.edu (MSS)

[†]These authors contributed equally to this work

**Competing interest:** The authors declare that no competing interests exist.

## Editor's evaluation

This paper addresses an important question: the relationship between the cell wall and other, primarily lipid, based components of the cell envelope. Building on previous work, the authors provide solid data suggesting that the activity of PonA2, a non-essential peptidoglycan synthase, promotes membrane partitioning through its role in cell wall synthesis. Altogether, this work provides valuable insight into the mechanisms coordinating the synthesis of separate layers of the bacterial cell envelope and as such should be of interest to microbiologists working on similar aspects of growth-related processes across bacterial systems.

## Introduction

Biological membranes are heterogeneous mixtures of lipids and proteins (***Bernardino de la Serna et al., 2016***; ***Singer and Nicolson, 1972***). In eukaryotic cells, membrane domains have been linked to diverse functions, including signal transduction, membrane sorting, protein processing, and virus trafficking (***Goñi, 2019***; ***Simons and Sampaio, 2011***). In bacterial cells, the molecular mechanisms and physiological significance of membrane partitioning in these organisms are only beginning to emerge. To date, the best-studied example is functional membrane microdomains (FMMs), which are present in

many bacterial species and are, like their eukaryotic counterparts, more liquid-ordered, or rigid, than the surrounding plasma membrane (*López and Kolter, 2010*). FMMs contribute to diverse biological functions, including signaling, cell morphology maintenance, and biofilm formation (*Bach and Bramkamp, 2013*; *Dempwolff et al., 2012a*; *Dempwolff et al., 2012b*; *Mielich-Süss et al., 2013*; *Yepes et al., 2012*; *Zielińska et al., 2020*). In contrast to FMMs, regions of increased fluidity (RIF) are more liquid-disordered than the surrounding membrane. RIFs are also present in many bacterial species but their function(s) remain relatively unexplored (*Gohrbandt et al., 2022*; *Molohon et al., 2016*; *Strahl et al., 2014*; *Wenzel et al., 2018*).

We have shown that bacteria of the pole-growing genus *Mycobacterium*—which includes both pathogens, such as *Mycobacterium tuberculosis,* and saprophytes, such as *Mycobacterium smegmatis*—partition their plasma membranes into the inner membrane domain (IMD; previously referred to as PMf or intracellular membrane domain) and conventional plasma membrane, which is termed the PM-CW as it is tightly associated with the cell wall (*García-Heredia et al., 2021*; *Hayashi et al., 2018*; *Hayashi et al., 2016*; *Morita et al., 2005*; *Puffal et al., 2022*). The biophysical nature of the IMD is not known but may be more liquid-disordered than the surrounding PM-CW given that the IMD is enriched in enzymes that act on membrane-fluidizing polyprenols and lipid intermediates carrying polymethylated fatty acids (*García-Heredia et al., 2021*; *Hayashi et al., 2018*; *Hayashi et al., 2016*; *Janas et al., 1994*; *Morita et al., 2005*; *Puffal et al., 2022*; *Schroeder et al., 1987*; *Valtersson et al., 1985*; *Vigo et al., 1984*; *Wang et al., 2008*). Biosynthetic pathways that are partitioned across the IMD and PM-CW include both the well-conserved, for example, cell wall peptidoglycan and plasma membrane phosphatidylethanolamine (PE), and the mycobacteria-specific, for example, plasma membrane glycolipids phosphatidylinositol mannosides (PIMs) and outer membrane phthiocerol dimycocerosate (PDIM) (*García-Heredia et al., 2021*; *Hayashi et al., 2018*; *Morita et al., 2005*; *Puffal et al., 2022*; *Puffal et al., 2018*). While the functional role of partitioning has not yet been demonstrated for most of these pathways, perturbations that departition the membrane also dampen peptidoglycan precursor production (*García-Heredia et al., 2021*), indicating that lateral partitioning may promote membrane-bound enzymatic reactions.

Partitioning in model membranes can be controlled by both bilayer-intrinsic factors, that is, protein (*Yuan et al., 2021*) and lipid composition (*Beales et al., 2005*; *Korlach et al., 1999*; *Veatch and Keller, 2003*), as well as bilayer-extrinsic factors, including temperature and interactions with other membranes and/or solid supports (*Gordon et al., 2008*; *Subramaniam et al., 2013*). It has been experimentally challenging to perform analogous experiments in the more-complex membranes of living cells (*Gohrbandt et al., 2022*). Therefore, most work on cellular membrane partitioning has concentrated on defining maintenance factors. One factor that maintains the behavior of membrane components across different domains of life is physical connection to extracellular surfaces. For example, enzymatic removal of neuronal extracellular matrix (*Frischknecht et al., 2009*) or the plant or bacterial cell wall (*Daněk et al., 2020*; *Feraru et al., 2011*; *Martinière et al., 2012*; *McKenna et al., 2019*; *Wagner et al., 2020*) can increase the lateral mobility of membrane proteins that are normally enriched in domains. However, even these loss-of-partitioning experiments can be complicated to interpret. Continuing the cell wall example, genetic or pharmacological inhibition of cell wall synthesis sometimes, but does not always, disrupt membrane partitioning or proxies thereof (*Daněk et al., 2020*; *Feraru et al., 2011*; *García-Heredia et al., 2021*; *Hayashi et al., 2018*; *Wagner et al., 2020*). As well, construction of the external cell wall depends on lipid-linked intermediates that reside in the plasma membrane, blurring the distinction between bilayer-intrinsic and -extrinsic factors that control membrane partitioning. More broadly, the molecular mechanisms by which cells partition their membranes are poorly understood, making it difficult to parse the physiological significance of partitioning.

While studying membrane departitioning can illuminate requirements for maintenance, tracking repartitioning can shed light on the requirements for de novo establishment. Here we develop a benzyl alcohol-induced membrane departitioning/repartitioning model to screen for factors that control partitioning in cells. We identify *ponA2* as a gene that helps *M. smegmatis* to recover from benzyl alcohol. PonA2 is a bifunctional cell wall synthase that is not required for *M. smegmatis* or *M. tuberculosis* growth but counteracts various stresses (*Kieser et al., 2015a*; *Li et al., 2022*; *Patru and Pavelka, 2010*; *Vandal et al., 2009a*; *Vandal et al., 2008*). Post-benzyl alcohol, PonA2 accelerates *M. smegmatis* regrowth and membrane repartitioning. Unlike its roles in localizing cell wall synthesis,

which depend on conserved transglycosylase and transpeptidase domains, or its role in maintaining cell morphology, which does not depend on either domain, PonA2's contribution to membrane repartitioning depends solely on its conserved transglycosylase domain. While the cell wall glycan backbone helps to maintain membrane partitioning, our data suggest that the role of PonA2 is specific to de novo partitioning and occurs at least in part via active cell wall polymerization. By establishing a simple in vivo model system for membrane departitioning and repartitioning, we start to untangle the intricate relationship between membrane and cell wall organization.

## Results

### Reversible departitioning and repartitioning of the mycobacterial plasma membrane

We previously demonstrated that the known plasma membrane fluidizer benzyl alcohol (*Friedlander et al., 1987*; *Ingram, 1976*; *Konopásek et al., 2000*; *Nagy et al., 2007*; *Strahl et al., 2014*; *Zielińska et al., 2020*) disrupts the association of peptidoglycan precursor synthase MurG to the *M. smegmatis* IMD and halts polar growth of the organism (*García-Heredia et al., 2021*). The effects of benzyl alcohol on peptidoglycan synthesis and growth reverse within 30 min after removing the chemical from the growth medium (*García-Heredia et al., 2021*). We wondered whether the ability of MurG to reassociate with the IMD post-benzyl alcohol extended to additional membrane domain constituents and/or fluidizers. Accordingly, we first treated *M. smegmatis* expressing functional fluorescent protein fusions to IMD-associated proteins GlfT2, Ppm1, and MurG (*García-Heredia et al., 2021*; *Hayashi et al., 2016*) with either benzyl alcohol or a different plasma membrane fluidizer, dibucaine (*Kinoshita et al., 2019*). IMD-associated proteins normally localize adjacent to the sites of polar growth in mycobacteria (*García-Heredia et al., 2021*; *Hayashi et al., 2016*). We found that Ppm1 and MurG were delocalized from the subpolar region by either chemical, whereas GlfT2 was delocalized only by dibucaine (*Figure 1A and C* and *Figure 1—figure supplement 1*). After 12 hr of recovery in the absence of the chemicals, the IMD marker proteins relocalized to their subpolar positions (*Figure 1B*). These data suggest that IMD-associated proteins can delocalize and relocalize to the subpolar region of the cell respectively in response to, and recovery from, different membrane-acting chemicals. The divergent behavior of Ppm1 and MurG vs. GlfT2 in the presence of benzyl alcohol also implies that the nature and/or strength of IMD association with constituent proteins may vary.

As a complementary way to track membrane constituents, we examined the IMD and PM-CW biochemically. Under normal growth conditions, the IMD can be separated from the PM-CW by sucrose density gradient fractionation (*García-Heredia et al., 2021*; *Hayashi et al., 2018*; *Hayashi et al., 2016*; *Morita et al., 2005*); the IMD fractions are less dense and distinct from the PM-CW fractions. As before (*García-Heredia et al., 2021*), we found that benzyl alcohol treatment resulted in the apparent loss of membranous material from the IMD fractions (*Figure 1D*). We also observed IMD recovery within the 3 hr post-benzyl alcohol washout. Taken together, these data suggest that benzyl alcohol treatment and washout can serve as a model for reversible membrane departitioning and repartitioning.

### Identification of non-essential genes that promote tolerance to and/or recovery from membrane fluidization

Subpolar IMD localization correlates closely with polar cell growth (*Hayashi et al., 2018*). We hypothesized that the ability of *M. smegmatis* to recover from benzyl alcohol depends at least in part on its ability to reform the IMD, and therefore, that genes that promote membrane partitioning would constitute a subset of the genes that enable *M. smegmatis* to tolerate and/or recover from benzyl alcohol. In mycobacteria, the essential, tropomyosin-like protein Wag31 (DivIVA) promotes polar cell wall assembly and rod shape and promotes IMD protein localization and isolability (*García-Heredia et al., 2018*; *Habibi Arejan et al., 2022*; *Jani et al., 2010*; *Kang et al., 2008*; *Melzer et al., 2018*). While DivIVA depletion slowed *M. smegmatis* growth, as expected, we observed only a small additional growth defect when bacteria were exposed to benzyl alcohol (*Figure 2—figure supplement 1*). We reasoned that DivIVA may contribute more to membrane partitioning maintenance than to initiation. Additionally, or alternatively, the essential role(s) of the protein may partially obscure its function in IMD biogenesis. Therefore, we used transposon sequencing (*Gawronski et al., 2009*; *Goodman*

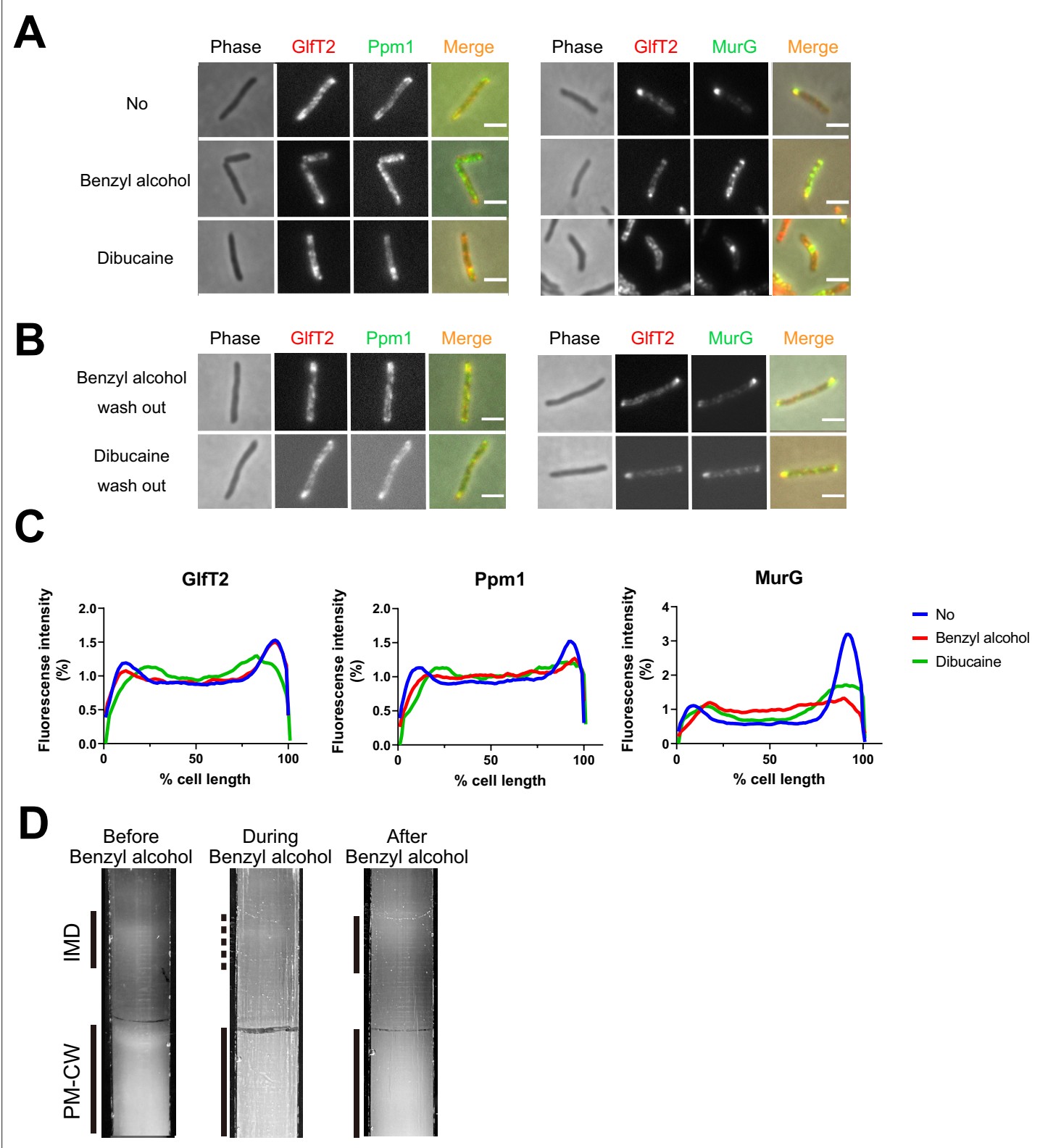

**Figure 1.** Delocalization of inner membrane domain (IMD) proteins by membrane-disrupting chemicals in *M. smegmatis*. mCherry-GlfT2, Ppm1-mNeonGreen, and MurG-Dendra2 are functional fluorescent protein fusions to well-established, IMD-associated proteins (*Hayashi et al., 2016*; *Hayashi et al., 2018*; *García-Heredia et al., 2021*). IMD proteins were imaged after 1 hr benzyl alcohol or 1 hr dibucaine treatment, (**A**), and again 12 hr after washout, (**B**). Pictures are representative of three independent experiments. Scale bars, 2.5 µm. (**C**) Fluorescence distributions of the fusion

*Figure 1 continued on next page*

Figure 1 continued

proteins after chemical treatment were calculated from three independent experiments. Lines show the average of all cells (50 < n < 75). Signal was normalized to cell length and total fluorescence intensity. Cells were oriented such that the brighter poles are on the right-hand side of the graph. See 'Materials and methods' for details. (D) Lysates from *M. smegmatis* taken before, during, and 3 hr after benzyl alcohol treatment were sedimented in sucrose density gradients and imaged.

The online version of this article includes the following source data and figure supplement(s) for figure 1:

**Source data 1.** Datasets for plot profiles of inner membrane domain (IMD) protein localizations.

**Figure supplement 1.** The localization of inner membrane domain (IMD) proteins was not changed by Dimethyl sulfoxide (DMSO) in *M. smegmatis* (compare to *Figure 1C*).

**Figure supplement 1—source data 1.** Dataset for plot profile.

*et al., 2009*; *Langridge et al., 2009*; *van Opijnen et al., 2009*) to search for non-essential factors that promote tolerance to and/or recovery from membrane disruption. Using the TRANSIT platform (*DeJesus et al., 2015*), we identified six candidate genes (*Figure 2* and *Table 1*) that fit our desired profile, that is, had fewer transposon insertions 16–24 hr post-benzyl alcohol exposure compared to DMSO vehicle control. Of these genes, *ponA2* showed the greatest statistical significance.

PonA2 is one of three bifunctional transglycosylase/transpeptidase enzymes, also known as class A penicillin-binding proteins (aPBPs), involved in cell wall peptidoglycan biosynthesis in *M. smegmatis*. PonA1 and likely PonA2 are enriched in the PM-CW (*García-Heredia et al., 2021*; *Hayashi et al., 2016*). While *M. smegmatis* PonA1 is essential for growth and/or viability, PonA2 and PonA3 are not (*Kieser et al., 2015a*; *Patru and Pavelka, 2010*; *Vandal et al., 2008*). PonA3 is not present in *M. tuberculosis* and not expressed in *M. smegmatis* under normal growth conditions (*Patru and Pavelka, 2010*). In contrast, PonA2 is conserved in both species and promotes survival of *M. smegmatis* under stress conditions such as starvation or oxygen depletion; *M. tuberculosis* tolerance to heat, some antibiotics, acid, reactive oxygen and nitrogen; and *M. tuberculosis* survival in some mouse backgrounds (*DeJesus et al., 2017*; *Kieser et al., 2015a*; *Li et al., 2022*; *Patru and Pavelka, 2010*; *Smith et al., 2022*; *Vandal et al., 2009a*; *Vandal et al., 2008*). As enzymatic removal of the *M. smegmatis* cell wall departitions the membrane (*García-Heredia et al., 2021*), we reasoned that PonA2 may promote membrane partitioning and opted to analyze this hit further.

## PonA2 contributes to efficient mycobacterial growth following benzyl alcohol exposure

To test whether PonA2 contributes to benzyl alcohol recovery, we constructed a clean deletion mutant (Δ*ponA2*) and a complemented strain (c*ponA2*) by introducing *ponA2* under its native promoter in the L5 integration site. We found that the number of colony-forming units (CFUs) for Δ*ponA2* was comparable to that of wild-type (*Figure 3A* and *Figure 3—figure supplement 1*) immediately after benzyl alcohol treatment, but that Δ*ponA2* growth lagged during the early post-washout recovery period (*Figure 3B*). The Δ*ponA2* growth delay was specific to benzyl alcohol as there was no delay after treatment with the DMSO vehicle control. Furthermore, the defect in benzyl alcohol recovery was restored in the complemented strain (c*ponA2*), indicating that the defect is due to the lack of *ponA2*. These data suggest that PonA2 helps *M. smegmatis* to recover from benzyl alcohol.

## PonA2 restores membrane partitioning after benzyl alcohol treatment

*M. smegmatis* eventually restores membrane partitioning after benzyl alcohol treatment (*García-Heredia et al., 2021*; *Figure 1B and D*), but the factors that promote repartitioning are unknown. We used subpolar enrichment of IMD-associated Ppm1 (*Figure 1*; *Hayashi et al., 2016*) as a readout for membrane partitioning before and after benzyl alcohol treatment. Prior to benzyl alcohol exposure, Ppm1 was enriched in the subpolar regions of wild-type, Δ*ponA2,* and the complemented mutant (*Figure 4A–C*). Immediately after benzyl alcohol exposure, subpolar enrichment diminished for all three strains. During the recovery period, Ppm1 relocalized to the subpolar regions within ~3 hr for wild-type and ~1 hr for the complement (*Figure 4C*). However, Ppm1 did not completely relocalize in Δ*ponA2* even up to 6 hr of outgrowth (*Figure 4A, C and D*). These data indicate that PonA2 contributes to spatial repartitioning of the plasma membrane following benzyl alcohol-induced fluidization.

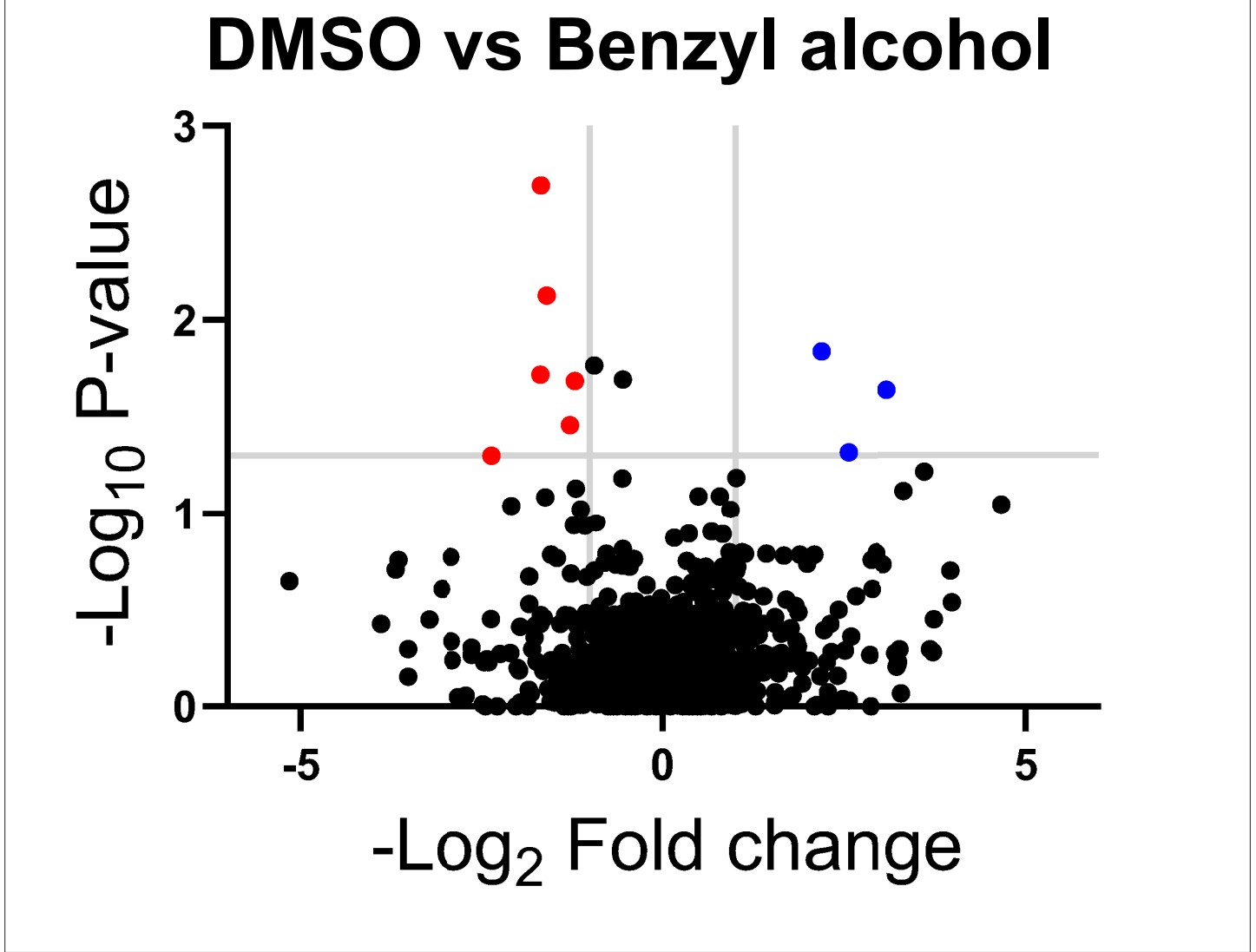

**Figure 2.** Genes in which transposon insertions are under (red, left) or over (blue, right) represented in *M. smegmatis* exposed to benzyl alcohol relative to DMSO vehicle control. A transposon library was treated with benzyl alcohol or DMSO for 1 hr. Benzyl alcohol was washed away and bacteria were resuspended in Middlebrook 7H9 growth medium. The $OD_{600}$ was then adjusted to 0.01 and bacteria were incubated for an additional 16–24 hr to an $OD_{600}$ of ~1.0. The library was then collected for DNA sequencing. Transposon insertion counts presented relative to counts ratio (treated/control) per gene and the corresponding p-values calculated by Mann–Whitney *U*-test (y-axis) from n = 3 independent experiments. The horizontal gray line indicates p<0.05; the vertical gray lines indicate two-fold change.

The online version of this article includes the following source data and figure supplement(s) for figure 2:

**Source data 1.** Raw analysis data of Tnseq.

**Figure supplement 1.** Modest contribution of DivIVA to *M. smegmatis* recovery from benzyl alcohol.

**Figure supplement 1—source data 1.** Dataset for resazurin growth curve.

As a complementary way to track membrane repartitioning, we examined the IMD and PM-CW by sucrose density gradient fractionation (*García-Heredia et al., 2021*; *Hayashi et al., 2018*; *Hayashi et al., 2016*; *Morita et al., 2005*). Benzyl alcohol treatment of both wild-type and Δ*ponA2 M. smegmatis* results in the apparent loss of membranous material from the IMD fractions (*Figure 4D*; *García-Heredia et al., 2021*). In contrast to wild-type, however, the mutant failed to recover the IMD within the 3 hr, post-benzyl alcohol washout period. This experiment suggests that, in addition to spatial repartitioning, PonA2 contributes to biochemical repartitioning of the plasma membrane after benzyl alcohol treatment.

**Table 1.** Genes identified by Tn-seq.

**Underrepresented in benzyl alcohol-treated *M. smegmatis* (candidates for promoting survival)**

| Gene locus | Gene name | Gene description |
|---|---|---|
| MSMEG_0846c | – | Putative monovalent cation/H⁺ antiporter subunit D |
| MSMEG_2775 | *nhaA* | Na⁺/H⁺ antiporter NhaA |
| MSMEG_4533c | – | Sulfate-binding protein |
| MSMEG_5488c | – | DNA-binding response regulator |
| MSMEG_5781c | *pstC* | Phosphate ABC transporter, permease protein PstC |
| MSMEG_6201 | *ponA2* | Bifunctional transglycosylase/transpeptidase |

**Overrepresented in benzyl alcohol-treated *M. smegmatis* (candidates for impairing survival)**

| MSMEG_2768 | – | OB-fold nucleic acid binding domain-containing protein |
|---|---|---|
| MSMEG_2772 | – | Amino acid permease |
| MSMEG_5694 | – | Hypothetical protein MSMEG_5694 |

We did not identify other enzymes involved in peptidoglycan biosynthesis from the Tn-seq analysis. Transposon insertions do not always interrupt gene function, and trans-complementation can occur when mutants are pooled. When tested individually, however, loss of other, non-essential cell wall synthases, including the monofunctional SEDS family transglycosylase RodA (Δ*rodA*) and multiple L,D-transpeptidases (Δ*ldtABE*) had no effect on *M. smegmatis* outgrowth post-benzyl alcohol (*Figure 4—figure supplement 1*). These data suggest that promotion of membrane repartitioning is not a universal property of peptidoglycan biosynthetic and/or remodeling enzymes.

## PonA2 does not localize membrane–cell wall interactions under unstressed conditions

Physical interactions between the plasma membrane and bilayer-extrinsic polymers have been proposed to maintain membrane partitioning in other systems (*Daněk et al., 2020*; *Feraru et al., 2011*; *Martinière et al., 2012*; *McKenna et al., 2019*; *Wagner et al., 2020*). In mycobacteria, we have demonstrated co-fractionation of the plasma membrane and cell wall upon mechanical cell lysis, that is, PM-CW (*García-Heredia et al., 2021*; *Hayashi et al., 2018*; *Hayashi et al., 2016*; *Morita et al., 2005*; *Puffal et al., 2022*). The existence of the PM-CW implies that the plasma membrane and cell wall are physically associated. To test this hypothesis further, we developed a microfluidics-based assay (*Figure 5A and B*) to quantify and visualize membrane-cell wall interactions in *M. smegmatis*. In *Escherichia coli*, hyperosmotic shock causes severe plasmolysis, whereby the plasma membrane retracts from the cell wall, indicating that these structures are not strongly associated (*Rojas et al., 2018*). As plasmolysis occurs in areas with weak membrane–cell wall association, we predicted that plasmolysis bays in *M. smegmatis* would preferentially form at sites of IMD enrichment, for example, subpolar foci. We exposed *M. smegmatis* in a microfluidics chamber to hyperosmotic shock and measured the number and location of plasmolysis bays. Bay formation was significantly more likely to occur in the subpolar region of the cell compared to the polar region (*Figure 5C*; note that the midcell region is a mix of IMD and PM-CW; *García-Heredia et al., 2021*; *Hayashi et al., 2018*; *Hayashi et al., 2016*; *Prithviraj et al., 2023*), supporting the notion that membrane–cell wall interaction is weaker in the IMD than in the PM-CW.

One potential mechanism by which PonA2 contributes to membrane partitioning is by localizing membrane–cell wall interactions to the PM-CW. Consistent with delocalized membrane–cell wall interactions, Δ*ponA2* was less likely to plasmolyse than wild-type *M. smegmatis* (*Figure 5D*). As in *M. tuberculosis* (*Kieser et al., 2015a*), the mutant is wider than wild-type (*Figure 5—figure supplement 1A*) but of similar length (*Figure 5E*), suggesting that decreased membrane surface area is unlikely to explain plasmolysis resistance in the absence of PonA2. Despite Δ*ponA2's* decreased propensity to plasmolyse, however, we did not detect differences in the subcellular distribution of plasmolysis in the mutant relative to wild-type (*Figure 5C*). These data suggest that PonA2 does not significantly

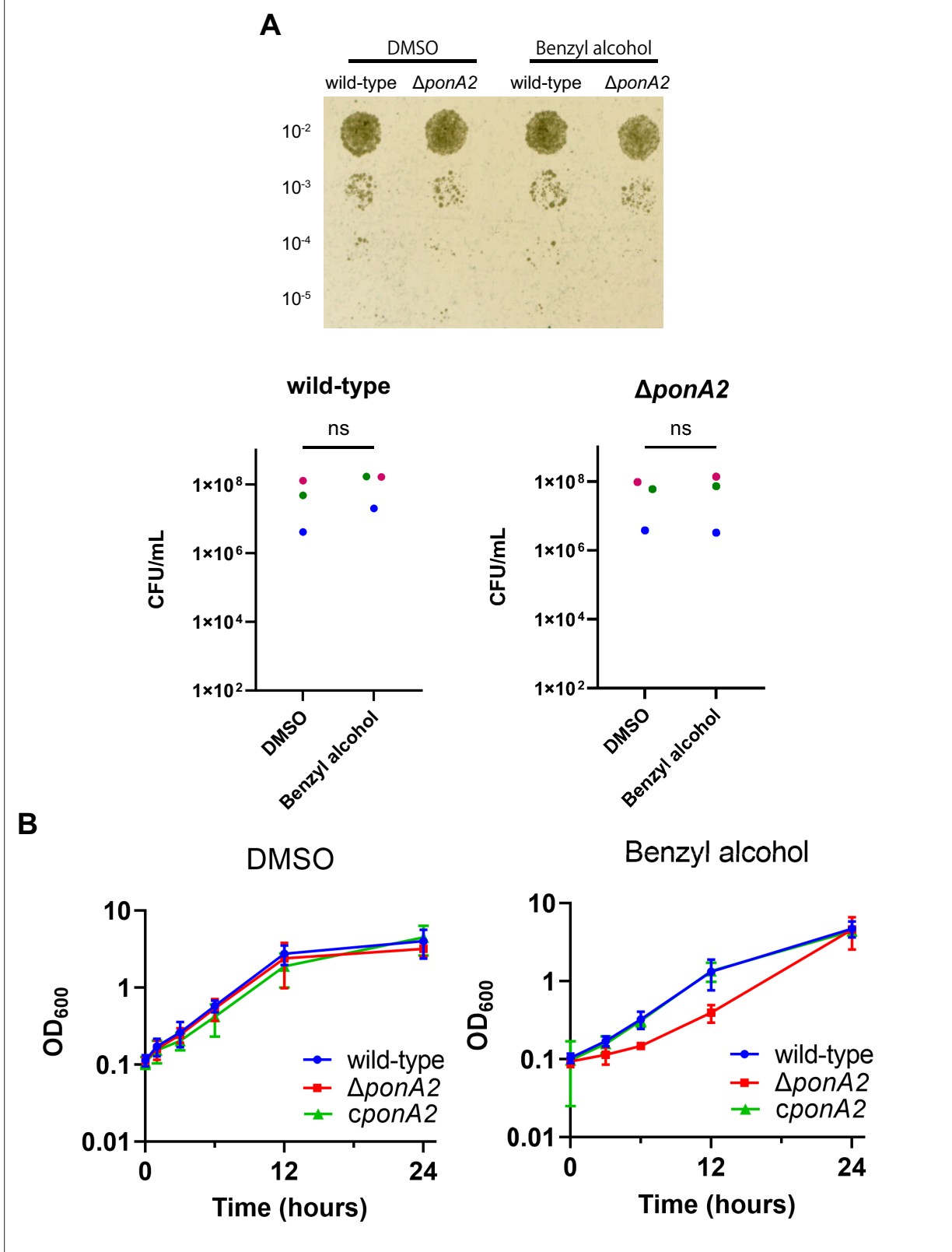

**Figure 3.** PonA2 contributes growth recovery from benzyl alcohol. (**A**) Top, wild-type or Δ*ponA2 M. smegmatis* were treated with benzyl alcohol for 1 hr, then 10-fold serial dilutions were spotted on Middlebrook 7H10 agar. The image is representative of three independent experiments. Bottom, colony-forming units (CFUs) were calculated from three biological replicates. Colors correspond to same-day replicates. ns, no statistically significant difference by Mann–Whitney *U*-test. p=0.4 or p>0.99 respectively. (**B**) Benzyl alcohol- or DMSO vehicle-treated wild-type, Δ*ponA2*, or complemented

*Figure 3 continued on next page*

*Figure 3 continued*

strain (c*ponA2*) were washed three times then grown in Middlebrook 7H9 medium. Lines show the average and SD obtained from three independent experiments.

The online version of this article includes the following source data and figure supplement(s) for figure 3:

**Source data 1.** Colony-forming unit (CFU) and OD$_{600}$ value for growth curves.

**Figure supplement 1.** PonA2 is dispensable for survival during benzyl alcohol treatment.

contribute to membrane partitioning by localizing membrane–cell wall interactions, at least under basal conditions.

## PonA2 promotes membrane repartitioning and regrowth post-fluidization via its conserved transglycosylase domain

Given that PonA2 is a bifunctional transpeptidase/transglycosylase, we hypothesized that one or both of its enzymatic activities contribute to membrane homeostasis. Accordingly, we made point mutations to alter well-conserved, catalytically active amino acids (*Figure 6—figure supplement 1*), substitutions that have been previously shown to eliminate activity in in vitro assays for PBP1a (*Born et al., 2006*) and PBP1b (*Terrak et al., 1999*) in *E. coli*, and the transpeptidase function of PonA1 in *M. smegmatis* (*Kieser et al., 2015b*). Specifically, we complemented Δ*ponA2* with *ponA2* alleles that bear E108T (transglycosylase inactive [TG-]), S405A (transpeptidase inactive [TP-]), or both (TG-/TP-) mutations. Complementation with wild-type *ponA2* restored the ability of *M. smegmatis* to tolerate a non-lethal challenge with moenomycin (*Melzer et al., 2022*), an antibiotic that targets aPBP transglycosylation (*Gampe et al., 2013*; *Meeske et al., 2016*; *Ostash and Walker, 2010*; *Welzel, 2005*), and to bind normally to the fluorescent β-lactam Bocillin-FL, which covalently labels active PBP peptidases (*Wissel and Weiss, 2004*). By contrast, and consistent with the notion that the TG- and TP- mutations respectively inactivate PonA2 transglycosylase and transpeptidase domains, these alleles failed to complement in the moenomycin and Bocillin-FL assays (*Figure 6—figure supplement 2*). Δ*ponA2* complemented with the TP- *ponA2* allele behaved similarly to Δ*ponA2* complemented with wild-type *ponA2* for both post-benzyl alcohol outgrowth (*Figure 6A*) and membrane repartitioning (*Figure 6B and C*). However, the recovery of Δ*ponA2* complemented with the TG- or TP-/TG- *ponA2* allele was delayed in a manner comparable to uncomplemented Δ*ponA2*. These data suggest that PonA2 promotes membrane repartitioning and regrowth post-benzyl alcohol via its conserved transglycosylase domain.

## PonA2's roles in supporting polar cell wall elongation and rod morphology are genetically separable from its roles in membrane partitioning

We wondered whether PonA2 supports membrane homeostasis by localizing cell wall assembly. Previously we showed that the polarity of peptidoglycan synthesis decreases in the absence of RodA, a SEDS family transglycosylase, and upon treatment with aPBP transglycosylation inhibitor moenomycin (*Melzer et al., 2022*). To more directly assay the function of PonA2, we labeled nascent peptidoglycan in wild-type and Δ*ponA2* after a brief incubation in the presence of alkyne-D-alanine-D-alanine (alkDADA, also called EDA-DA; *García-Heredia et al., 2018*; *Liechti et al., 2014*) and detected the presence of the alkyne probe via copper-catalyzed alkyne-azide cycloaddition (CuAAC) to a fluorescent azide label. In wild-type and Δ*ponA2* mycobacteria, fluorescence was enriched at the cell poles, the sites of cell elongation in this genus (*Aldridge et al., 2012*; *Joyce et al., 2012*; *Kieser and Rubin, 2014*; *Meniche et al., 2014*; *Santi et al., 2013*; *Singh et al., 2013*; *Thanky et al., 2007*; *Figure 7A*). However, as with RodA absence or moenomycin treatment (*Melzer et al., 2022*), there was a modest decrease in the polarity of nascent peptidoglycan in Δ*ponA2* compared to wild-type (*Figure 7B and C*). Moreover, a subpopulation of mutant cells had clear cell bulging (*Figure 7A* and *Figure 5—figure supplement 1A*), a phenotype that has long been linked to peptidoglycan defects (*Burdon, 1946*; *Chung et al., 2009*; *Hett et al., 2010*; *Huang et al., 2008*; *Typas et al., 2010*; *Vigouroux et al., 2020*). The morphological defects are consistent with prior observations of aberrant width and morphology, respectively, in Δ*ponA2 M. tuberculosis* (*Kieser et al., 2015a*) and stationary phase Δ*ponA2 M. smegmatis* (*Patru and Pavelka, 2010*). However, in contrast to the role

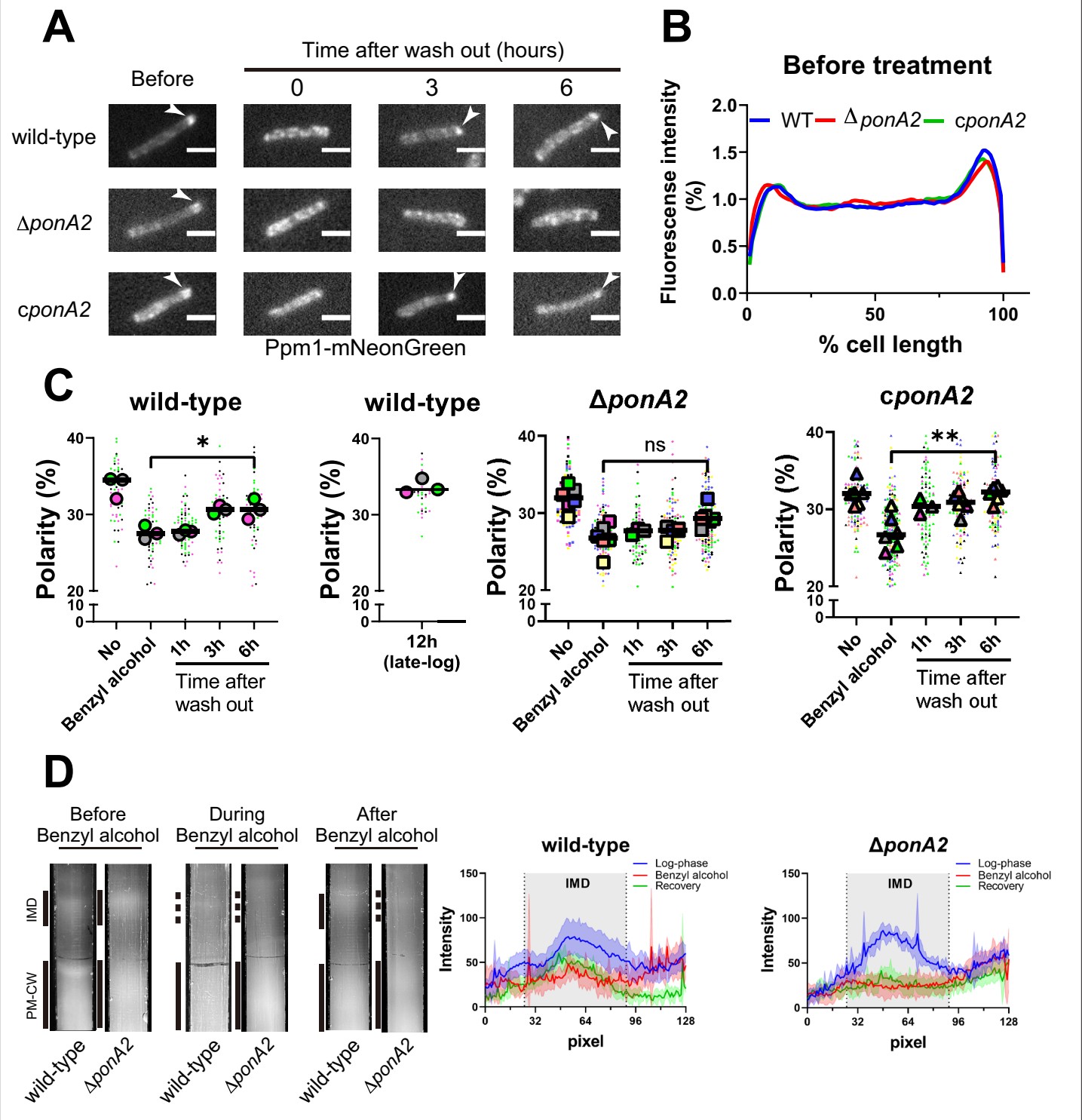

**Figure 4.** PonA2 promotes membrane repartitioning after benzyl alcohol treatment. (**A**) Fluorescence imaging of *M. smegmatis* expressing Ppm1-mNeonGreen before benzyl alcohol treatment or after benzyl alcohol washout. Arrowheads indicate subpolar foci of Ppm1-mNeonGreen. Scale bars, 2.5 μm. Pictures are representative of three independent experiments. (**B**) Fluorescence of cells imaged as in (**A**) were quantitated from three independent experiments as in *Figure 1C*. Lines show the average of all cells (50 < n < 69). (**C**) The percentage of signal associated with the distal 15% of rod-shaped cells is quantified to indicate polarity of fluorescence distribution. Each color in the super plots (*Lord et al., 2020*) represents an independent biological replicate. Smaller symbols are the polarities of each cell, and larger symbols are the means of the polarity in each replicate. Statistical significance was determined by the Kruskal–Wallis test, followed by Dunn's multiple-comparison test. ns, no statistically significant difference (p=0.3313); *p=0.0130; **p=0.0037. Data were obtained from three or six independent experiments. (**D**) Lysates from *M. smegmatis* at indicated

*Figure 4 continued on next page*

*Figure 4 continued*

time points were sedimented in a sucrose density gradient. Representative images of the collection tubes after sucrose gradient fractionation at left. Densitometry of the membranous material (highlighted next to the tubes with solid and dashed gray lines) shown in the right panel. The lines are the average pixel values derived from three distinct lines across images obtained from a representative experiment. The lighter areas are the standard deviations. Images of wild-type sucrose gradient fractionations are repeated from *Figure 1D* for clarity.

The online version of this article includes the following source data and figure supplement(s) for figure 4:

**Source data 1.** Raw values for plot profiles of inner membrane domain (IMD) protein, raw values for the super plots, and raw values of densitometry.

**Figure supplement 1.** Following exposure to benzyl alcohol or DMSO vehicle control, cells were washed, resuspended in Middlebrook 7H9, and incubated with 0.0015% of resazurin.

**Figure supplement 1—source data 1.** Dataset for resazurin growth curve of mutants.

of PonA2 in post-benzyl alcohol membrane repartitioning and regrowth, which depend solely on its transglycosylation domain (*Figure 6*), the role of PonA2 in localizing cell wall synthesis depended on both its transglycosylase and transpeptidase domains (*Figure 7D*). Furthermore, the role of PonA2 in maintaining normal rod shape did not depend on either domain (*Figure 5—figure supplement 1B*). The genetic separability of the phenotypes suggests that PonA2's contributions to membrane partitioning, polar peptidoglycan assembly, and cell morphology are distinct.

## PonA2 promotes tolerance to benzyl alcohol-induced membrane permeabilization

Given that loss of PonA2 sensitizes mycobacteria to a variety of stresses (*DeJesus et al., 2017*; *Kieser et al., 2015a*; *Li et al., 2022*; *Patru and Pavelka, 2010*; *Vandal et al., 2009b*; *Vandal et al., 2008*), we also considered the possibility that the membrane partitioning and growth phenotypes of Δ*ponA2* uncovered by benzyl alcohol were associated with enhanced susceptibility to the chemical. For example, high fluidity can enhance the permeability of model membranes (*Frallicciardi et al., 2022*; *Gabba et al., 2020*; *Lande et al., 1995*; *Rossignol et al., 1982*). To test membrane permeability, we incubated wild-type and Δ*ponA2* with propidium iodide, a dye that fluoresces upon DNA intercalation and is normally membrane-impermeant. At baseline, neither wild-type nor Δ*ponA2* stained appreciably with propidium iodide (*Figure 7—figure supplement 1A*). However, ~20% of wild-type cells were propidium iodide-positive after benzyl alcohol treatment (*Figure 7—figure supplement 1A and C*), indicating that benzyl alcohol-induced fluidization can compromise the membrane barrier. Under the same conditions, ~60% of the Δ*ponA2* cells were propidium iodide-positive (*Figure 7—figure supplement 1A and C*). These data suggest that Δ*ponA2* cells are more permeable than wild-type following benzyl alcohol exposure.

Propidium iodide staining is often used to detect dead cells. However, the CFUs of bacteria with or without PonA2 or benzyl alcohol were similar (*Figure 3A*), suggesting that enhanced permeability was not lethal. These data are consistent with a previous report showing that >50% of log-phase, propidium iodide-stained *Mycobacterium frederiksbergense* were culturable (*Shi et al., 2007*). We reasoned that live, propidium iodide-positive *M. smegmatis* may be able to grow upon dye washout, and that DNA synthesis and subsequent cell division would dilute the fluorescence over time. To test whether the propidium iodide-positive cells were alive, we examined the fluorescence of the propidium iodide-positive population after washing out the dye. As a negative control, we incubated heat-killed cells in growth medium and confirmed that there was no change in fluorescence over time. In contrast, the propidium iodide-positive populations of wild-type and Δ*ponA2* were reduced by half after 3 and 6 hr of outgrowth, respectively (*Figure 7—figure supplement 1B*), a rate of dilution that roughly correlated with the bulk population growth rate of the strains during the same time frame (*Figure 3B*). Thus, while loss of PonA2 exacerbates benzyl-alcohol-induced membrane permeabilization, the perturbations alone or combined do not compromise viability.

As with the delayed growth and membrane repartitioning phenotypes of benzyl alcohol-exposed Δ*ponA2*, enhanced membrane permeability was complemented by wild-type *ponA2* but not by the TG- or TP-/TG- *ponA2* alleles (*Figure 7—figure supplement 1C*). Complementation with the TP-*ponA2* allele was intermediate between that with wild-type and TG- *ponA2,* although not statistically significant. These phenotypes imply that the role of the PonA2 in membrane permeability and partitioning may be genetically linked. Therefore, we tested whether enhanced uptake of benzyl alcohol

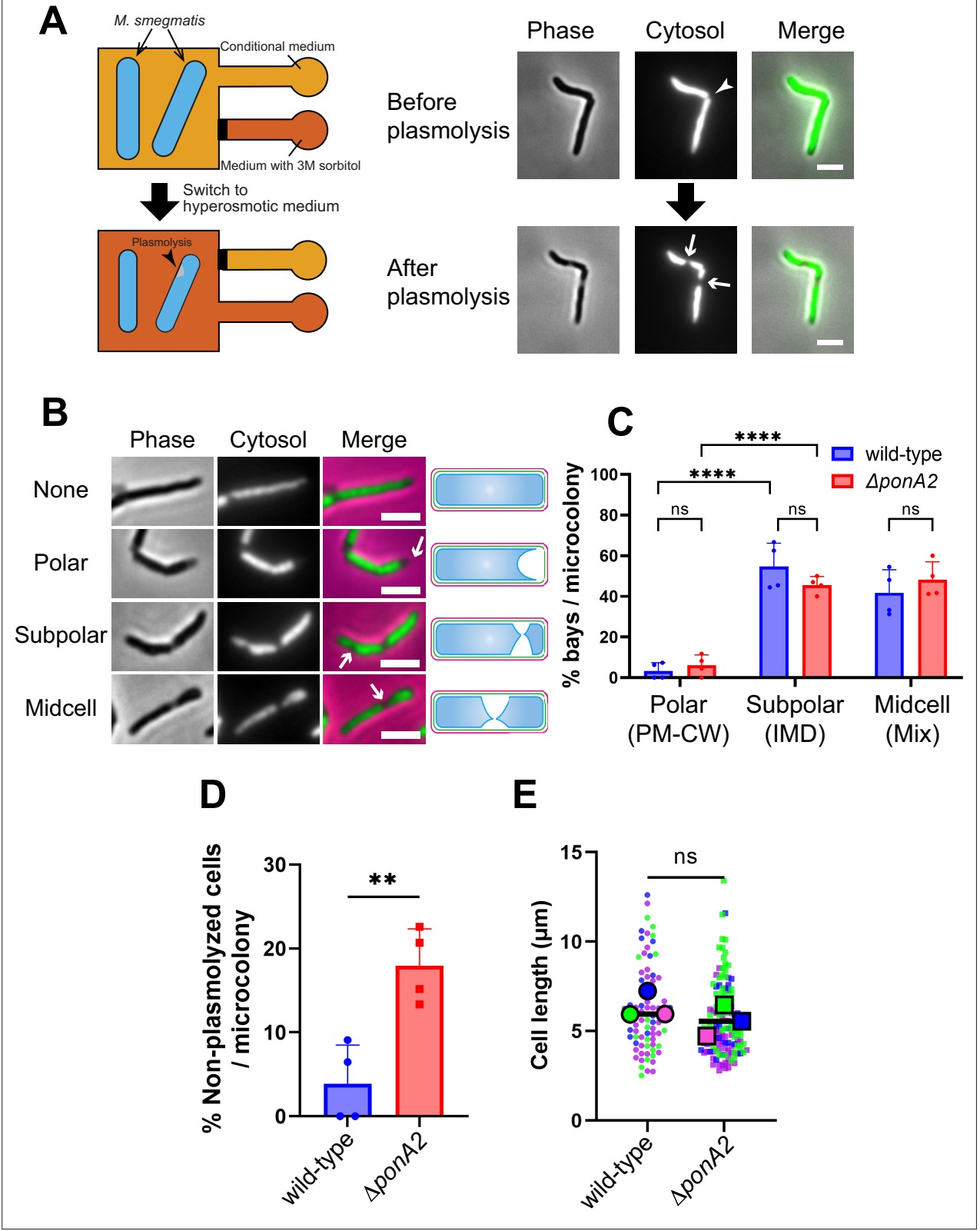

**Figure 5.** Membrane–cell wall interactions are weaker in subpolar regions. (**A**) Left, schematic of the microfluidics plasmolysis assay. GFP-expressing *M. smegmatis* were subjected to hyperosmotic shock (3 M sorbitol) in microfluidics device and imaged before, during, and after shock. Right, images before and after hyperosmotic shock. Cells were perfused with 7H9 medium for 5 min and then perfused with 7H9 + 3 M sorbitol for 5 min. Cells were imaged in phase (50 ms) and GFP (50 ms) every 30 s. An arrowhead indicates the septum while arrows indicate sites of plasmolysis. Scale bars, 2.5 μm.

*Figure 5 continued on next page*

*Figure 5 continued*

Images are representative of four independent experiments. (**B**) Cytoplasmic GFP enables visualization and quantitation of plasmolysis bays. Arrows indicate representative sites of plasmolysis. Scale bars, 2.5 µm. Images are representative of two independent experiments. (**C**) Analysis of the sites of plasmolysis (as defined in [**B**]) in wild-type (n = 85) and Δ*ponA2* (n = 101) *M. smegmatis* from two biological replicates performed in duplicate. For this analysis, we removed cells that did not plasmolyse. Statistical significance was determined by the two-way ANOVA test, followed by Šídák's multiple-comparisons test. ****$p<0.0001$, ns, $p>0.99$. Polar, subpolar, and midcell bays spatially correlate with the PM-CW, inner membrane domain (IMD), and a mixture of PM-CW and IMD, respectively (*García-Heredia et al., 2021*; *Hayashi et al., 2018*; *Hayashi et al., 2016*; *Prithviraj et al., 2023*). (**D**) The proportion of cells that did not plasmolyse in (**C**) was calculated for wild-type (n = 90) and Δ*ponA2* (n = 123) *M. smegmatis*. Statistical significance was determined by the Mann–Whitney *U*-test. **$p=0.045$. (**E**) Cell lengths were analyzed for wild-type (n = 84) and Δ*ponA2* (n = 132) *M. smegmatis* from three biological replicates. Each color in the super plots (*Lord et al., 2020*) represents an independent biological replicate. Smaller symbols are the lengths of individual cells, and larger symbols are the means of each replicate. Statistical significance was determined by the Mann–Whitney *U*-test. ns, $p=0.4$.

The online version of this article includes the following source data and figure supplement(s) for figure 5:

**Source data 1.** Raw values of the proportions of where plasmolysis bay were seen, population without plasmolysis, and cell length.

**Figure supplement 1.** *M. smegmatis* width profiles of cells along the normalized cell length were determined by Oufti (*Nguyen et al., 2007*; *Paintdakhi et al., 2016*) and a Python script here (copy archived at *Sparks, 2023*).

could explain the delayed growth and membrane repartitioning of Δ*ponA2* relative to wild-type (*Figures 3B, 4 and 6*). Accurate measurement of benzyl alcohol cell accumulation is challenging given its volatile nature. However, we reasoned that an uptake defect should manifest as concentration-dependent, differential sensitivity to benzyl alcohol between wild-type and Δ*ponA2*, an outcome that we did not observe (*Figure 3—figure supplement 1*). In aggregate, our data are most consistent with the notion that the PonA2 transglycosylase domain promotes tolerance to benzyl alcohol-induced membrane permeabilization.

## The cell wall glycan backbone contributes to membrane partitioning maintenance

One potential mechanism by which PonA2 transglycosylation promotes membrane partitioning is via cell wall integrity. While we previously showed that spheroplasting delocalizes multiple IMD markers (*García-Heredia et al., 2021*), complete removal of the cell wall may have pleiotropic effects on membrane physiology. More recently, we showed that limited peptidoglycan digestion by the glyco-side hydrolases lysozyme and mutanolysin delocalizes the IMD-enriched protein MurG (*Melzer et al., 2022*). To test the specificity of this observation, we examined the distribution of two additional IMD markers, Ppm1 and GlfT2 (*Hayashi et al., 2018*; *Hayashi et al., 2016*). Ppm1, but not GlfT2, modestly delocalized upon lysozyme and mutanolysin treatment (*Figure 8A* and *Figure 8—figure supplement 1*). These results were consistent with the divergent behavior of Ppm1 and MurG vs. GlfT2 in the presence of benzyl alcohol (*Figure 1A and C*) and suggest that the maintenance of membrane partitioning is supported, directly or indirectly, by the cell wall glycan backbone.

De novo membrane partitioning is supported by concurrent cell wall polymerization. PonA2 helps *M. smegmatis* to tolerate (*Figure 7—figure supplement 1*) and recover from (*Figures 3, 4 and 6*) benzyl alcohol-induced membrane disruption. In the absence of benzyl alcohol treatment, however, Δ*ponA2* has normal growth, impermeability and Ppm1 localization, as well as a biochemically isolable IMD (*Figures 3, 4 and 6*, *Figure 7—figure supplement 1*). While Δ*ponA2* was overall less likely to plasmolyse compared to wild-type (*Figure 5D*), there were no obvious differences in the subcellular distribution of plasmolysis between the strains (*Figure 5C*). These data indicate that PonA2 is dispensable for maintaining membrane partitioning under unstressed conditions, at least to the limit of our detection. Therefore, we focused on understanding the role of PonA2 in de novo membrane partitioning.

Efficient membrane repartitioning post-benzyl alcohol depends on the conserved transglycosylase domain of PonA2 (*Figure 6B and C*). We considered the possibility that active cell wall polymerization might accelerate membrane partitioning. We previously demonstrated that treatment of *M. smegmatis* with the cell wall-targeting antibiotic D-cycloserine delocalizes IMD markers, but that it takes approximately one generation and the IMD remains biochemically isolable throughout (*Hayashi et al., 2018*). In other organisms, cell wall polymerization and cell wall expansion can temporarily continue in the presence of D-cycloserine (*Pisabarro et al., 1986*; *Sugimoto et al., 2017*). To investigate a

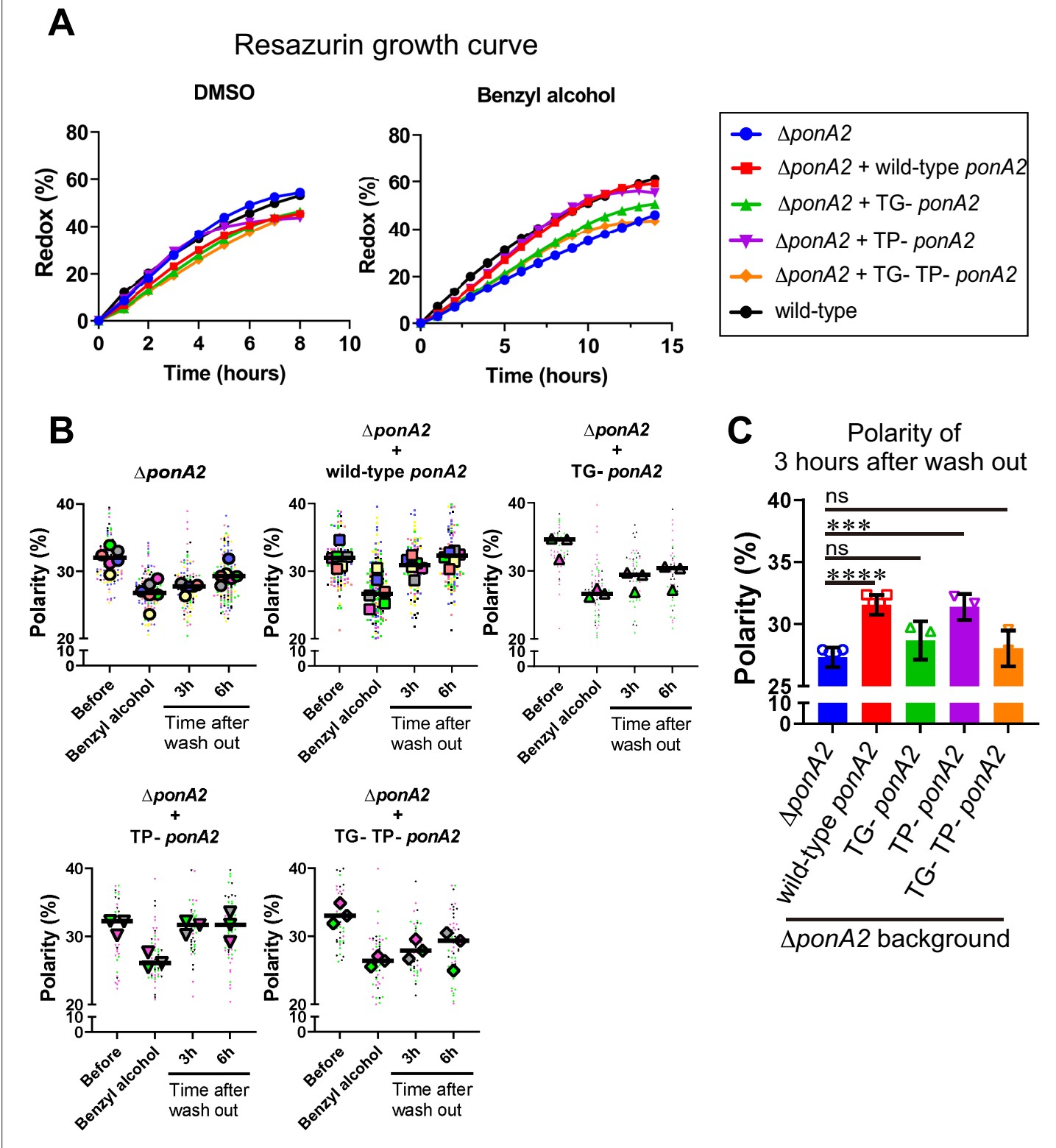

**Figure 6.** Transglycosylase domain of PonA2 accelerates regrowth and membrane repartitioning post-benzyl alcohol. (**A**) Following exposure to benzyl alcohol or the DMSO vehicle control, cells were washed, resuspended in Middlebrook 7H9, and incubated with 0.0015% of resazurin. Data were obtained from three independent experiments and are means of biological duplicates or triplicates. (**B**) Inner membrane domain (IMD) marker (Ppm1-mNeonGreen) polarities were assessed in mutants before, during, or 3- or 6 hr after benzyl alcohol treatment as in *Figure 4C*. Data from Δ*ponA2* and Δ*ponA2* complemented with wild-type *ponA2* are repeated from *Figure 4C* for ease of comparison. Each color in the super plots (*Lord et al., 2020*)

*Figure 6 continued on next page*

*Figure 6 continued*

represents an independent biological replicate. Smaller symbols are the polarities of each cell, and larger symbols are the means of each replicate. (**C**) The polarity of 3 hr time points from (**B**) were compiled and compared across mutants. The bar graph shows the average and standard deviation. Statistical significance was determined by the Kruskal–Wallis test, followed by Dunn's multiple-comparison test. ns, no statistically significant difference (p=0.2538 or 0.7601 respectively); ****p<0.0001; ***p=0.0002.

The online version of this article includes the following source data and figure supplement(s) for figure 6:

**Source data 1.** Raw values of resazurin growth curve, super plots, and polarity of 3 hr after wash out.

**Figure supplement 1.** Protein alignments of PBP1a of *E. coli* and PonA1 and PonA2 of *M. smegmatis.*

**Figure supplement 2.** Enzymatic activity of catalytic inactive mutants.

**Figure supplement 2—source data 1.** Dataset for resazurin growth curve of moenomycin-treated mutants.

role for cell wall polymerization in de novo membrane partitioning, we first titrated D-cycloserine and the aPBP transglycosylase inhibitor moenomycin to identify concentrations that were sublethal but inhibitory. We next compared *M. smegmatis* growth in the presence or absence of the antibiotics during the recovery period after benzyl alcohol washout. D-cycloserine inhibited growth equally in *M. smegmatis* exposed or not to benzyl alcohol, at all concentrations tested (*Figure 8B*, left). By contrast, moenomycin inhibited growth to a modestly greater extent in *M. smegmatis* that had been previously exposed to benzyl alcohol (*Figure 8B*, right). These results, together with our genetic data that demonstrate a role for PonA2's transglycosylase domain in recovery from benzyl alcohol (*Figure 6*), suggest that cell wall polymerization accelerates de novo membrane partitioning.

## Discussion

Lateral organization is likely to be a key regulator of plasma membrane function yet is experimentally challenging to manipulate in living cells. Given that benzyl alcohol can reversibly fluidize membranes in other bacteria and in eukaryotic cells (*Balogh et al., 2005*; *Chabanel et al., 1985*; *Coster and Laver, 1986*; *Friedlander et al., 1987*; *Hubbell et al., 1970*; *Konopásek et al., 2000*; *Nagy et al., 2007*; *Paterson et al., 1972*; *Shigapova et al., 2005*; *Strahl et al., 2014*; *Zielińska et al., 2020*), our inducible departitioning/repartitioning model (*Figure 1*) and subsequent screening (*Figure 2*) may be a generalizable approach to genetically dissect the mechanisms by which cellular membranes are partitioned.

Factors that establish and maintain plasma membrane partitioning are likely important for cell fitness but may be distinct from each other. We previously demonstrated that there is a close correlation between membrane partitioning and cell growth in *M. smegmatis* (*Hayashi et al., 2018*) and that the known chemical fluidizer benzyl alcohol departitions the membrane and halts growth in this organism (*García-Heredia et al., 2021*). Building on these observations, we screened here for mycobacterial genes that promote recovery from benzyl alcohol, a subset of which we hypothesized would also promote membrane partitioning. We identified a factor, cell wall synthase PonA2, that establishes membrane partitioning via its transglycosylase domain. Our earlier work had suggested that peptidoglycan damage and/or removal (*García-Heredia et al., 2021*; *Melzer et al., 2022*), but not specific defects in peptidoglycan synthesis (*Hayashi et al., 2018*), departition the *M. smegmatis* membrane. Similar, seemingly incongruous observations have been made for the roles of the cell wall and its synthesis in other bacteria (*Wagner et al., 2020*) and in plants (*Daněk et al., 2020*). By dissecting the role of PonA2 in the presence or absence of benzyl alcohol-induced membrane disruption, we find that active cell wall polymerization helps in establishing membrane partitioning, which in turn is maintained at least in part by the completed cell wall polymer. Membrane partitioning correlates with active *M. smegmatis* growth, although a causative role of partitioning in growth remains speculative.

We envision at least three scenarios, not mutually exclusive, by which PonA2 regulates membrane partitioning. Membrane-bound proteins and/or other biomolecules may partition the bilayer by tethering it to the cell wall and influencing protein and lipid diffusion. In this first model (*Figure 9A*), PonA2 accelerates membrane partitioning by fashioning a cell wall structure that is conducive to tethering. In the second model (*Figure 9B*), PonA2 accelerates membrane partitioning because nascent cell wall polymers themselves act as transient tethers between the plasma membrane (via a polyprenol phosphate anchor) and the cell wall (via partial incorporation into the existing peptidoglycan mesh).

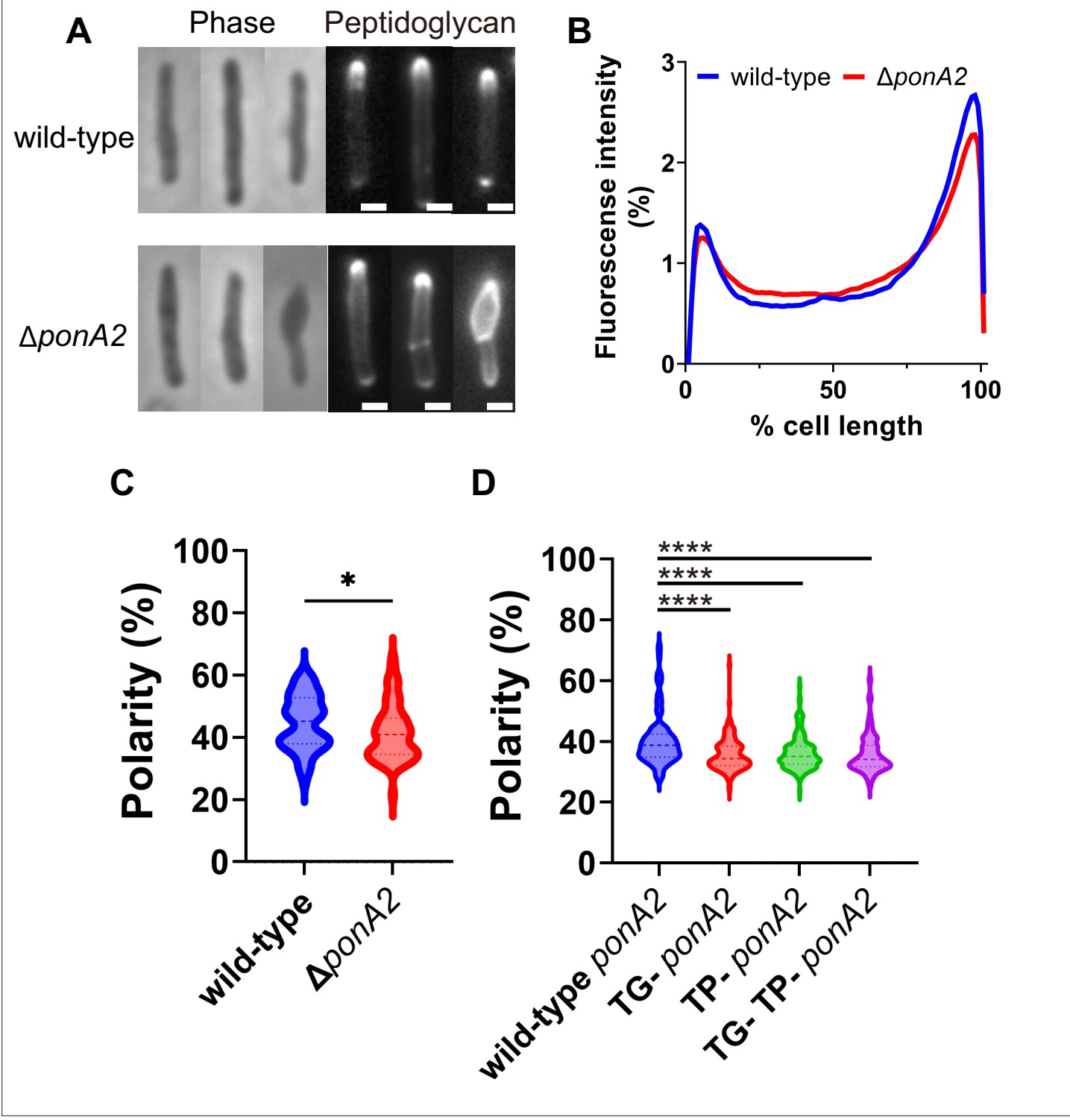

**Figure 7.** PonA2 localizes peptidoglycan synthesis and maintains cell morphology. (**A**) Nascent peptidoglycan in wild-type or Δ*ponA2 M. smegmatis* was labeled for 15 min (~10% generation) with alkyne-ᴅ-alanine-ᴅ-alanine (alkDADA) followed by copper-catalyzed alkyne-azide cycloaddition (CuAAC) with AFDye488 Azide. Scale bars, 1 μm. Pictures are representative of three independent experiments. (**B**) Wild-type and Δ*ponA2* strains were labeled as in (**A**) and subcellular fluorescence was quantitated as in *Figure 1C*. Lines show the average of all cells (50 < n < 68) obtained by three independent experiments. (**C**) The percentage of signal associated with the distal 15% of rod-shaped cells quantified to indicate polarity of fluorescence distribution. Statistical significance was determined by Mann–Whitney *U*-test. p-value, *p=0.0227. (**D**) Polarity ratio of catalytically inactive mutants. Statistical

*Figure 7 continued on next page*

*Figure 7 continued*

significance was determined by the Kruskal–Wallis test, followed by Dunn's multiple-comparison test. Data were obtained from the three independent experiments. ****p<0.0001.

The online version of this article includes the following source data and figure supplement(s) for figure 7:

**Source data 1.** Raw values of plot profiles and polarity of peptidoglycan synthesis.

**Figure supplement 1.** PonA2 promotes cell impermeability upon benzyl alcohol treatment.

**Figure supplement 1—source data 1.** Dataset for propidium-iodide-positive population.

In the third model (*Figure 9C*), PonA2 accelerates membrane partitioning by controlling lipid II pools. In in vitro membrane systems, lipid II both generates and homes to more fluid regions (*Ganchev et al., 2006*; *Jia et al., 2011*; *Valtersson et al., 1985*). Spatial and/or temporal regulation of lipid II consumption in cellular membranes may serve a bilayer-intrinsic role in membrane partitioning. Finally, while we favor a direct role for cell wall synthesis and/or structure in organizing the plasma membrane as the most parsimonious explanation for our data, it is possible that PonA2 indirectly supports other aspects of the cell envelope, which in turn feed back on the plasma membrane.

In laterally growing, rod-shaped bacteria, it is thought that the SEDS-family transglycosylase RodA lays the template for cell wall elongation while the bifunctional, transglycosylase/transpeptidase aPBPs fill in the gaps for maintenance and repair (*Cho et al., 2016*; *Mueller et al., 2019*; *Murphy et al., 2021*; *Paradis-Bleau et al., 2010*; *Typas et al., 2010*; *Vigouroux et al., 2020*). Unlike most organisms in which this model has been tested, pole-growing *Mycobacteriales* lack the cytoskeletal protein MreB and do not require RodA for viability or shape (*Arora et al., 2018*) so the division of labor has been less clear (*Melzer et al., 2022*; *Sher et al., 2021*). Given that PonA2 but not RodA contributes to membrane partitioning in *M. smegmatis* (*Figure 4—figure supplement 1*), and the models above (*Figure 9*), it may be that PonA2 and RodA build nascent cell wall polymers and/or mature cell wall structures with different tethering capacities. Additionally, or alternatively, PonA2 and RodA may consume lipid II in ways that differentially impact the fluidity of the membrane.

PonA2 is not required for *M. smegmatis* or *M. tuberculosis* growth. However, the bifunctional transglycosylase/transpeptidase protects these organisms from various stresses, including antibiotics with different structures and cellular targets (*Kieser et al., 2015a*; *Li et al., 2022*; *Patru and Pavelka, 2010*; *Vandal et al., 2009a*; *Vandal et al., 2008*), a phenotype that suggests enhanced small molecule permeability. Aberrant peptidoglycan synthesis in the absence of PonA2 may disrupt the cell envelope layers outside of the peptidoglycan, including the mycomembrane. As mycomembrane-disrupting mutations can sensitize mycobacteria to many antibiotics (*Gao et al., 2003*; *Li et al., 2022*; *Liu and Nikaido, 1999*; *Nguyen et al., 2005*; *Philalay et al., 2004*; *Vilchèze et al., 2014*), it is often assumed that the mycomembrane is the primary determinant of mycobacterial impermeability and intrinsic antibiotic resistance. Our work suggests that *ponA2* mutations can also impact the organization and integrity of the layer inside of the peptidoglycan, the plasma membrane. It is an open question whether and how plasma membrane defects contribute to the stress-specific phenotypes of mycobacterial *ponA2* mutants.

Individual steps of the bacterial cell wall synthesis pathway are partitioned within the plasma membrane. For example, RIFs are enriched for MurG, the synthase for the polyprenol phosphate-linked cell wall precursor lipid II (*Müller et al., 2016*; *Strahl et al., 2014*). FMMs are enriched for lipid II flippase MurJ and for extracellular synthases that use lipid II to assemble the cell wall (*García-Fernández et al., 2017*). In mycobacteria, we have demonstrated that lipid II is made in the IMD, then trafficked to, and likely polymerized in, the PM-CW (*García-Heredia et al., 2021*). Our data now support a model in which active cell wall synthesis helps to initiate a positive feedback loop between

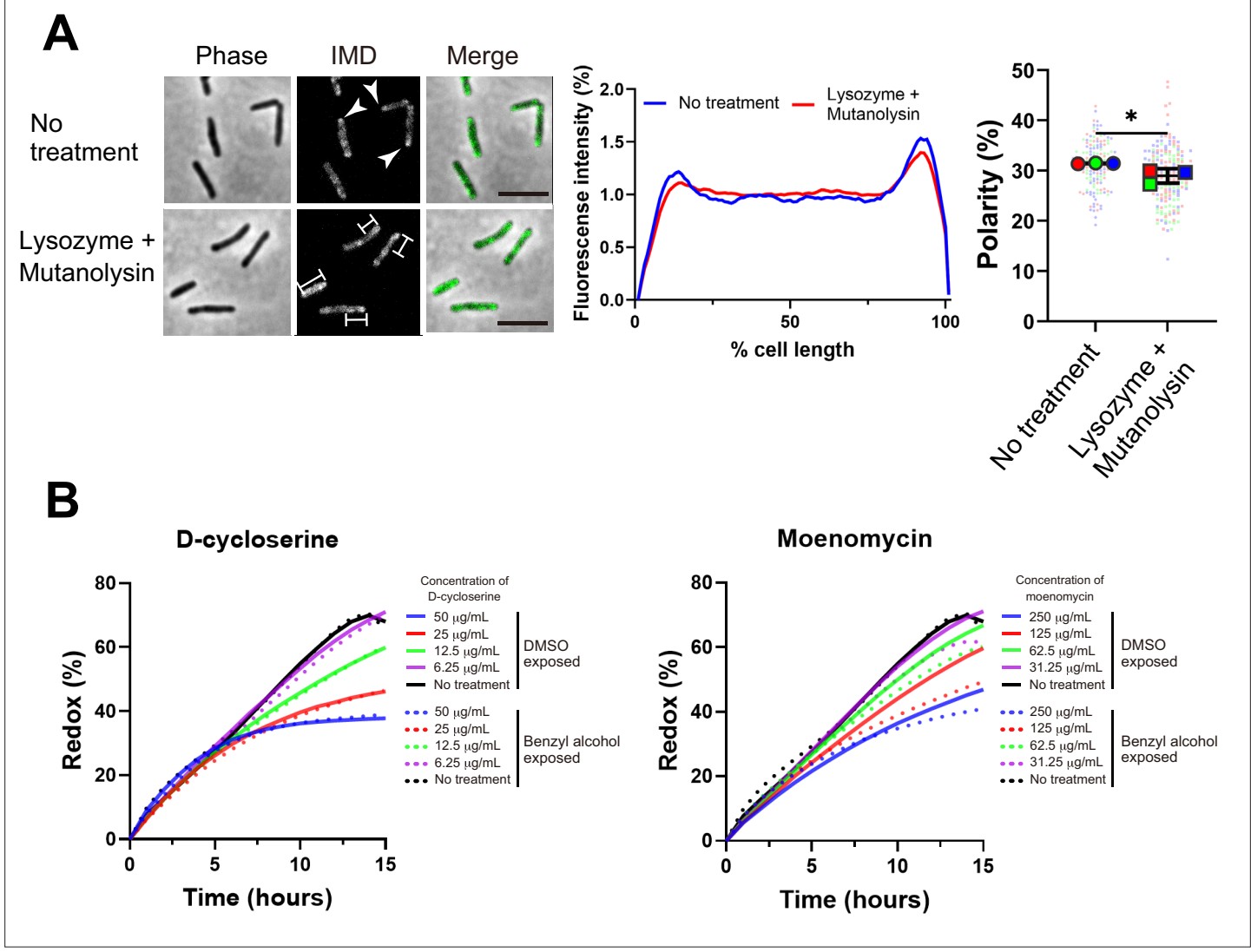

**Figure 8.** Maintenance and establishment of membrane partitioning are respectively supported by the cell wall glycan backbone and active cell wall polymerization. (**A**) Left, images of *M. smegmatis* expressing inner membrane domain (IMD) marker Ppm1-mNeonGreen ± 60 min treatment with cell wall hydrolases lysozyme and mutanolysin. Arrowheads indicate subpolar foci of Ppm1-mNeonGreen. Bars highlight dispersed fluorescent signal of Ppm1-mNeonGreen. Scale bars, 5 μm. Middle, quantitation of Ppm1-mNeonGreen polarity for cells with no treatment (n = 134) or lysozyme/mutanolysin treatment (n = 187). Lines show the average of all cells. Right, the percentage of Ppm1-mNeonGreen signal associated with the distal 15% of rod-shaped cells is quantified to indicate polarity of fluorescence distribution. Each color in the super plots (*Lord et al., 2020*) represents an independent biological replicate. Smaller symbols are the polarities of each cell and larger symbols are the means of each replicate. The line shows the average and standard deviation. Data were obtained from three independent experiments. Mann–Whitney *U* p-value, *p=0.04. (**B**) Bacteria treated with benzyl alcohol (dashed line) or DMSO vehicle (solid line) were washed then grown in Middlebrook 7H9 with D-cycloserine or moenomycin at the indicated concentrations. The lines are the average of two or three biological replicates.

The online version of this article includes the following source data and figure supplement(s) for figure 8:

**Source data 1.** Raw values of plot profiles and resazurin growth curves.

**Figure supplement 1.** Top, images of *M. smegmatis* expressing inner membrane domain (IMD) marker mCherry-GlfT2 ±60 min treatment with cell wall hydrolases lysozyme and mutanolysin.

**Figure supplement 1—source data 1.** Dataset for plot profiles of inner membrane domain (IMD) protein and super plot of polarity.

membrane and cell wall organization, which is then maintained by the cell wall glycan backbone. Damage to the cell wall, which we accomplish experimentally by treatment with glycoside hydrolases, interrupts this pro-growth feedback loop and departitions the membrane (*Figure 8A*; *García-Heredia et al., 2021*; *Melzer et al., 2022*), which in turn delocalizes the synthesis of peptidoglycan and other

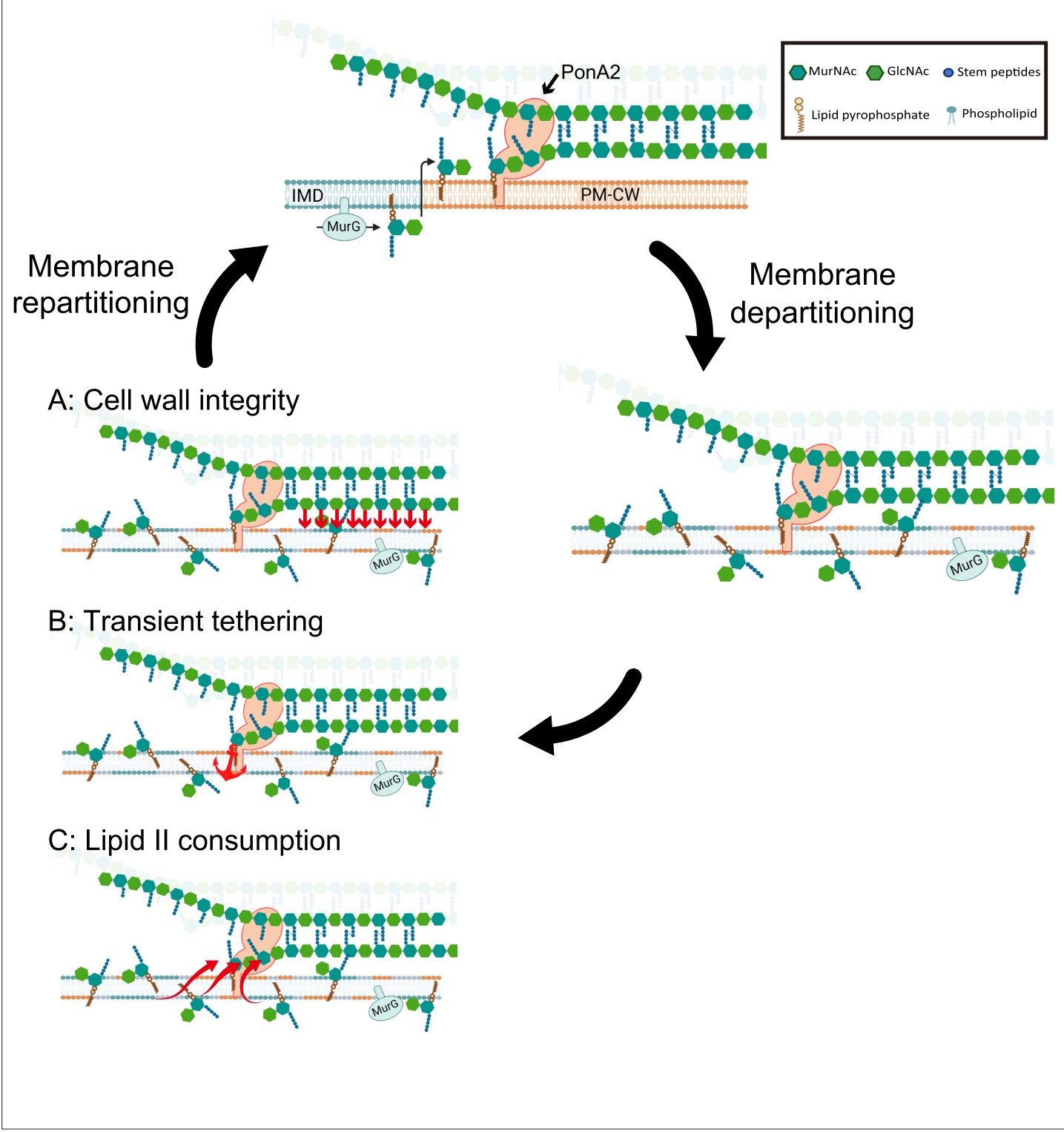

**Figure 9.** Working models for the role PonA2 in de novo plasma membrane partitioning. Inner membrane domain (IMD) (blue) and PM-CW (orange) have distinct proteomes and lipidomes (*Hayashi et al., 2016*). PonA2, likely enriched in the PM-CW (*García-Heredia et al., 2021*; *Hayashi et al., 2016*), polymerizes peptidoglycan using lipid II as a donor substrate, which in turn is produced by MurG in the IMD. PonA2 also cross-links nascent glycan strands into the existing cell wall. Upon membrane fluidization, lipid II and IMD-enriched proteins are no longer confined to the IMD (*García-Heredia et al., 2021*). PonA2 may promote membrane repartitioning post-fluidization via: (**A**) a mature peptidoglycan structure that directly or indirectly interacts with the membrane (red arrows), (**B**) nascent peptidoglycan polymers that transiently tether the membrane to the cell wall, and/or (**C**) consumption of lipid II, in turn regulating membrane fluidity.

cell envelope components to the sidewall (*García-Heredia et al., 2021*; *García-Heredia et al., 2018*). We hypothesize that delocalized envelope synthesis enables mycobacteria to robustly respond to and repair cell-wide damage (*García-Heredia et al., 2018*). Once the stress has passed, we further posit that the pro-growth, membrane–cell wall feedback loop begins anew. The interplay between membrane and cell wall organization may enable mycobacterial cells to adjust polar growth and side-wall repair as needed.

# Materials and methods

**Key resources table**

| Reagent type (species) or resource | Designation | Source or reference | Identifiers | Additional information |
|---|---|---|---|---|
| Strain, strain background (*Mycobacterium smegmatis* mc²155) | *M. smegmatis* | NC_008596 in GenBank | | Wild-type *M. smegmatis* |
| Strain, strain background (*M. smegmatis*) | mCherry-GlfT2 and Ppm1-mNeonGreen | *Hayashi et al., 2016* | | See reference for details |
| Strain, strain background (*M. smegmatis*) | mCherry-GlfT2 and MurG-Dendra2 | This study | | The mutant was generated as described in Supplementary material and methods |
| Strain, strain background (*M. smegmatis*) | D*ponA2* wild-type background | This study | | The mutant was generated as described in Supplementary material and methods |
| Strain, strain background (*M. smegmatis*) | D*ponA2* wild-type background complementing wild-type *ponA2* | This study | | The mutant was generated as described in Supplementary material and methods |
| Strain, strain background (*M. smegmatis*) | D*ponA2* wild-type background complementing TG- *ponA2* | This study | | The mutant was generated as described in Supplementary material and methods |
| Strain, strain background (*M. smegmatis*) | D*ponA2* wild-type background complementing TP- *ponA2* | This study | | The mutant was generated as described in Supplementary material and methods |
| Strain, strain background (*M. smegmatis*) | D*ponA2* wild-type background complementing TG- and TP- *ponA2* | This study | | The mutant was generated as described in Supplementary material and methods |
| Strain, strain background (*M. smegmatis*) | D*ponA2* mC-GlfT2 and Ppm1-mNG background | This study | | The mutant was generated as described in Supplementary material and methods |
| Strain, strain background (*M. smegmatis*) | D*ponA2* mC-GlfT2 and Ppm1-mNG background complementing wild-type *ponA2* | This study | | The mutant was generated as described in Supplementary material and methods |
| Strain, strain background (*M. smegmatis*) | D*ponA2* mC-GlfT2 and Ppm1-mNG background complementing TG- *ponA2* | This study | | The mutant was generated as described in Supplementary material and methods |
| Strain, strain background (*M. smegmatis*) | D*ponA2* mC-GlfT2 and Ppm1-mNG background complementing TP- *ponA2* | This study | | The mutant was generated as described in Supplementary material and methods |
| Strain, strain background (*M. smegmatis*) | D*ponA2* mC-GlfT2 and Ppm1-mNG background complementing TG- and TP- *ponA2* | This study | | The mutant was generated as described in Supplementary material and methods |
| Strain, strain background (*M. smegmatis*) | mCherry-GlfT2/DivIVA-eGFP-ID | *García-Heredia et al., 2021* | | See reference for details |
| Strain, strain background (*M. smegmatis*) | GFP expressing wild-type | This study | | The mutant was generated as described in Supplementary material and methods |
| Strain, strain background (*M. smegmatis*) | GFP expressing D*ponA2* background of wild-type | This study | | The mutant was generated as described in Supplementary material and methods |
| Software, algorithm | MATLAB codes | *García-Heredia et al., 2018*; *Paintdakhi et al., 2016* | RRID:SCR_001622 | Scripts designed for MATLAB to analyze the fluorescence profiles along a cell body from data collected in Oufti. |
| Other | Python script | This study | | See 'Materials and methods' and *Source code 1* for details |
| Software, algorithm | ImageJ | *Schindelin et al., 2012* | RRID:SCR_003070 | See reference for details |
| Software, algorithm | GraphPad Prism 9 | Commercially available | RRID:SCR_002798 | |

## Bacterial strains and growth conditions

Markerless, knock-in *M. smegmatis* strains expressing both HA-mCherry-GlfT2 and Ppm1-mNeonGreen-cMyc or HA-mCherry-GlfT2 alone were previously established (*Hayashi et al., 2016*). *M. smegmatis* mc²155 (wild-type), Δ*ponA2*, and Δ*ponA2 L5::ponA2* (wild-type and various alleles

of *ponA2*) were grown in Middlebrook 7H9 growth medium (BD Difco, Franklin Lakes, NJ) supplemented with 11 mM glucose, 14.5 mM NaCl, 0.4% (vol/vol) glycerol, and 0.05% (vol/vol) Tween-80 (Sigma–Aldrich, St. Louis, MO), as well as kanamycin (50 µg/mL) and/or hygromycin (50 µg/mL) where appropriate. Bacteria were grown at 37°C with shaking at 130 rpm. For chemical treatments, 200 µL of 5 M benzyl alcohol in DMSO (Sigma-Aldrich) or 20 µL of 0.2 M dibucaine in water (Sigma-Aldrich) was added to a 10 mL log-phase culture to achieve a final concentration of 100 mM or 0.4 mM, respectively,. The same volume of DMSO (200 µL to 10 mL culture, 2% [v/v]) or water (20 µL for 10 mL culture) was added as negative control. Phosphate-buffered saline (PBS) with 0.05% (vol/vol) Tween-80 (PBST) was used to wash out benzyl alcohol prior to resuspending bacteria in Middlebrook 7H9.

## Transposon library construction

A transposon library of *M. smegmatis* was made by using Himar mutagenesis as previously described (*Long et al., 2015*; *Siegrist and Rubin, 2009*). Briefly, Φ MycoMarT7 phage (*Piddock et al., 2000*) and a log-phase *M. smegmatis* culture were mixed and incubated for at 37°C for 4 hr. Cells were spread on Middlebrook 7H10 medium supplemented with 11 mM glucose, 14.5 mM NaCl, 0.5% glycerol, 0.05% Tween 80, and 50 µg/mL of kanamycin and incubated for 2–3 d at 37°C, yielding a library of ~$10^5$ mutants. The library was prepared by scraping colonies and stored as frozen stocks in Middlebrook 7H9 medium with 25% (vol/vol) glycerol at –80°C for further experiments. Library coverage of TA dinucleotide sites was determined to be ~35% by Illumina sequencing.

## Benzyl alcohol selection of transposon libraries

A frozen stock was thawed and 20 µL of the stock was inoculated to 20 mL of Middlebrook 7H9 medium. After overnight incubation at 37°C to allow the library to recover, this library starter culture was subcultured into 100 mL cultures to make a log-phase culture. The log-phase culture was treated by benzyl alcohol or DMSO vehicle control for 1 hr. Both cultures were washed three times with PBST and resuspended in Middlebrook 7H9 at a starting $OD_{600}$ of 0.01. The cultures were incubated at 37°C until the $OD_{600}$ became 1.0 in order to standardize the number of outgrowth generations between libraries to approximately 6.5 generations.

## Sequencing of transposon mutant libraries

Genomic DNA was extracted from benzyl alcohol- or DMSO-treated transposon libraries, and the library mutant composition was determined by sequencing amplicons of the transposon-genome junctions as previously described using primers indicated in *Supplementary file 1* (*DeJesus et al., 2017*; *Long et al., 2015*). On average, library sequencing yielded between 0.5 million and 4 million unique transposon-inserted-sequences which cover over 35% of the possible TA sites in the genome.

## Mapping and quantification of transposon insertions

Raw sequence data were processed using the TPP tool from the TRANSIT TnSeq analysis platform (*DeJesus et al., 2015*), and transposon genome junctions were mapped to the *M. smegmatis* mc²155 reference genome (GenBank accession number NC_018143.1) using the Burroughs-Wheeler aligner (*Li and Durbin, 2009*). To account for possible PCR amplification biases, reads corresponding to the same TA site and possessing the same 7-nucleotide barcode were derived from the same template, and these duplicate reads were discarded from the final template counts. Data in *Figure 2* were obtained from three biological replicates.

## Identification of genes affecting fitness under benzyl alcohol selection

Genes conditionally affecting fitness in the presence of benzyl alcohol were identified using the resampling test module in the TRANSIT analysis platform as previously described (*DeJesus et al., 2017*; *DeJesus et al., 2015*). In brief, we treated DMSO (a control treatment) and benzyl alcohol to transposon mutant library in triplicate. After washing out by PBST thrice, $OD_{600}$ was adjusted to 0.01. The cultures were incubated for 16–24 hr to $OD_{600}$~1.0. DNA was isolated from 30 mL of culture, sequenced, and analyzed as described in the previous work (*DeJesus et al., 2017*; *DeJesus et al., 2015*).

## Construction of plasmids and mutants

### pMUM264

To delete the endogenous *ponA2* gene, we amplified upstream and downstream regions of *ponA2* using the A980 and A981 (all primers are shown in **Supplementary file 1**). These two fragments were assembled into pCOM1 (**Hayashi et al., 2016**) at Van91I sites by Gibson assembly (New England Biolabs). The assembled plasmid, pMUM264, was transformed into *M. smegmatis* by electroporation, and positive clones were isolated based on hygromycin resistance and SacB-dependent sucrose sensitivity. Correct deletion of the *ponA2* gene was confirmed by PCR.

### pMUM280

Primers A980 and A981 were designed to amplify *ponA2*, including 192 bp of upstream native promoter region from wild-type. The PCR fragment was assembled into pMUM 126 (**Hayashi et al., 2016**) at KpnI-XbaI sites by Gibson assembly. The assembled plasmid, pMUM280, was transformed into *M. smegmatis* by electroporation, and positive clones were isolated based on kanamycin resistance.

### pMUM293 (TG-) and 294 (TP-)

Primers A995 to A998 were designed to make point mutations of *ponA2* as shown in **Supplementary file 1** using the Q5 Site-Directed Mutagenesis Kit (New England Biolabs). After mutations were confirmed by sequencing, the resulting plasmid, pMUM293 or 294, was transformed into *M. smegmatis* as above.

### pMUM295 (TG-/TP-)

The part of pMUM293 which includes the TG region was digested by SacI and MluI and the fragment was inserted into the same region of pMUM294 by ligation.

## CFUs and growth curves

Wildtype, Δ*ponA2*, and the complemented strain (c*ponA2*) cells were grown to stationary phase, then back-diluted and allowed to grow overnight to log phase (OD$_{600}$ 0.5–0.8). Cultures were treated with DMSO or benzyl alcohol (100 mM of final concentration) at 37°C with shaking at 150 rpm for 1 hr. The treated cultures were washed with PBST three times and resuspended in Middlebrook 7H9 medium at a starting OD$_{600}$ of 0.1 for continuous OD measurement in 125 mL flasks (**Figure 3C**) or 96-well-plate with antibiotics (**Figure 8B**). BioTek Synergy 2 was used for the growth curve in 96-well-plate. Aliquots (20 μL) were serial diluted with 7H9 media (200 μL) and 5 μL of aliquot is plated for CFUs.

## Resazurin growth curves

Our Tn-seq experiment (**Figure 2**) and initial validation (**Figure 3B**) were performed in 7H9 growth medium that lacked albumin. We found that *M. smegmatis* clumped when grown in the same medium in 96-well plate format, precluding accurate OD$_{600}$ readings. Addition of albumin prevented clumping. However, genetic or chemical perturbation of PonA2 no longer had a phenotype post-benzyl alcohol when albumin was included in the growth medium. While we do not yet understand this effect, we speculate that albumin might mitigate benzyl alcohol-induced fluidization and/or alter osmolarity. Therefore, we used resazurin reduction as an alternative, high-throughput method to assess *M. smegmatis* growth ± benzyl alcohol for the experiments in **Figures 6A and 8B** (**Eagen et al., 2018**). In brief, cells were washed three times with PBST and resuspended in Middlebrook 7H9 following a 1 hr treatment with DMSO vehicle control or benzyl alcohol. Then, 200 μL of culture and 20 μL of 0.015% (w/v) resazurin (Acros Organics) were mixed in 96-well-plates, and absorbance at 570 nm and 600 nm wwas measured by the Synergy H1 Hybrid microplate reader (BioTek) overnight. The percent of reduced resazurin was calculated as before (**Eagen et al., 2018**).

## Microscopy and image analysis

An aliquot (5 μL) of bacterial culture was inoculated on an agar pad (1% agarose in water) placed on a glass slide glass. Images were acquired on Nikon Eclipse E600. Cell outlines were traced using Oufti (**Nguyen et al., 2007**; **Paintdakhi et al., 2016**). Intensity profiles were generated using MATLAB code as described in **García-Heredia et al., 2018**. Polarity ratios were calculated by combining signal

values for 15% of the cell length on either pole and dividing the sum by total cell fluorescence. Super plots were generated as described.

## Membrane fractionation

Log-phase *M. smegmatis* cells treated with benzyl alcohol were harvested by centrifugation and washed in PBST. One gram of wet pellet was resuspended in 4 mL of lysis buffer containing 25 mM HEPES (pH 7.4), 20% (wt/vol) sucrose, 2 mM EGTA, and a protease inhibitor cocktail (Thermo Fisher Scientific, Waltham, MA) as described (*Morita et al., 2005*). Bacteria were lysed by high pressure of nitrogen (~2000 ppm). The lysate was centrifuged, and supernatant was spotted on the top of a tube containing sucrose gradient (20–50% [w/v], 25 mM HEPES). The sample was sedimented by ultra centrifuge (Beckman-Coulter) at 35,000 rpm for 6 hr on SW-40Ti rotor (Beckman-Coulter) at 4°C as in the previous literature (*García-Heredia et al., 2021*; *García-Heredia et al., 2021*; *Hayashi et al., 2016*; *Morita et al., 2005*). The tubes were imaged after sedimentation.

## Densitometry analysis

Images in *Figure 4D* were converted to grayscale in ImageJ (*Schindelin et al., 2012*). Three parallel lines were drawn from the top of the tube to the middle of tube. The gray values along the lines were quantified in each pixel. The average and standard deviation were plotted.

## Bocillin-FL

Cell lysates were prepared by bead beating as in the previous literature (*Rahlwes et al., 2017*). Then, 30 µg protein of cell lysate was incubated with 100 pmol Bocillin-FL (Invitrogen) in a total volume of 7.5 µL for 30 min in 37°C (*Levine and Beatty, 2021*). The sample was mixed with 2.5 µL of 4×-SDS-loading buffer and boiled for 3 min at 98°C. The entire sample was subjected to SDS-PAGE analysis. Gel was imaged by Amersham ImageQuant 800 system (Cytiva).

## Cell wall damage

Cells expressing mCherry-GlfT2 were grown to stationary phase, then back-diluted and allowed to grow overnight to log phase ($OD_{600}$ = 0.5–0.8). Cultures were incubated at 37°C shaking at 300 rpm in Benchmark Scientific MultiTherm Shaker H5000-H for 1 hr with 500 µg/mL lysozyme (Sigma-Aldrich, prepared fresh) and 500 U/mL mutanolysin (Sigma-Aldrich). Cells were imaged as described above.

## Cell envelope labeling

AlkDADA was custom synthesized by WuXi Apptec. Mid-log *M. smegmatis* was labeled with 2 mM alkDADA for 15 min. Cells were washed with PBST containing 0.01% BSA (PBSTB) and fixed in 2% formaldehyde at room temperature for 10 min. Cells were washed twice and applied for the reaction with CuAAC AFDye488 Azide (Click Chemistry Tool, Scottsdale, AZ) as described (*García-Heredia et al., 2018*; *Siegrist et al., 2013*).

## Cell width morphology profiles

Cells were placed on an agar pad slide, and imaged by phase microscopy (Nikon Eclipse E600, Nikon Eclipse Ti with ×100 objectives, N.A.=1.30). From the phase microscopy images, cells were outlined and segmented using Oufti (*Paintdakhi et al., 2016*). Cell width data were exported from Oufti and analyzed using a custom Python script. Using this script, cell lengths were normalized to a length of 1 (midcell = 0.5) and their width along their length was plotted as a line with each line representing a single cell. Multiple cell width profiles were superimposed on top of each other to visualize the major morphological trend (rod vs. blebbed). Additionally, percentages of cells with maximum widths ≥0.95 µm (green dotted line) were counted and the total percentage of cells obtaining widths at or above these thresholds was displayed.

## Imaging in microfluidic devices

We used a Nikon Eclipse Ti2-E inverted fluorescence microscope with a ×100 (N.A. 1.40) oil-immersion objective for imaging DU885 electron-multiplying charge-coupled device camera (Andor) for imaging (*Rojas et al., 2018*). Devices were kept at 37°C for imaging. Cells were streaked out on LB agar containing 50 µg/mL hygromycin and incubated at 37°C for 2–3 d. Single colonies were inoculated

into Middlebrook 7H9 containing 50 µg/mL hygromycin and incubated for ~48 hr at 37°C. The cells were then back diluted and added to the B04A microfluidic perfusion plate (CellASIC) during exponential phase. Plates were loaded with medium pre-warmed to 37°C. Cells were loaded into the plate, which was incubated at 37°C, without shaking, for 30 min before imaging. Medium was exchanged using the ONIX microfluidic platform (CellASIC). In the case where cells were stained with RADA, a TAMRA-based fluorescent D-amino acid (Tocris Bioscience), 1 µM RADA was added to the culture upon back dilution. If the cells were not stained with RADA, Alexa Fluor 647 NHS succinimidyl ester (Thermo Fisher Scientific) was added to the media as an occlusion dye (it is not cell wall permeable and thus can be used to track the cells). Cells were perfused with Middlebrook 7H9 medium for 5 min and then hyperosmotically shocked with 7H9 + 3 M sorbitol for 10 min.

## Acknowledgements

We thank Drs. Eric Rubin and Chidi Akusobi for Tn-seq guidance, Alam García-Heredia for advice, and Jungwoo Lee and Jun-Goo Kwak for plate reader guidance. Some of the data were obtained at the University of Massachusetts Flow Cytometry, Biophysical Characterization, and Genomics Resource Core Facilities, with support from the Institute for Applied Life Sciences and directors Drs. Amy Burnside, Stephen Eyles, and Ravi Ranjan. Research was supported by funds from the National Institutes of Health (NIH) under awards R21 AI144748 (YSM and MSS), DP2 AI138238 (MSS), R03 AI140259-01 (YSM), R35GM143057 (ERR), and T32 GM008515 (ESM, under the Chemistry and Biology Interface Program at the University of Massachusetts Amherst). TK was supported by a postdoctoral fellowship from Uehara Memorial Foundation.

## Additional information

### Funding

| Funder | Grant reference number | Author |
|---|---|---|
| National Institute of Allergy and Infectious Diseases | R21 AI144748 | Yasu S Morita M Sloan Siegrist |
| National Institute of Allergy and Infectious Diseases | DP2 AI138238 | M Sloan Siegrist |
| National Institute of Allergy and Infectious Diseases | R03 AI140259-01 | Yasu S Morita |
| National Institute of General Medical Sciences | R35GM143057 | Enrique R Rojas |
| National Institute of General Medical Sciences | T32 GM008515 | Emily S Melzer |
| Uehara Memorial Foundation | | Takehiro Kado |

The funders had no role in study design, data collection and interpretation, or the decision to submit the work for publication.

### Author contributions

Takehiro Kado, Conceptualization, Data curation, Formal analysis, Funding acquisition, Investigation, Visualization, Methodology, Writing - original draft, Project administration, Writing – review and editing; Zarina Akbary, Data curation, Formal analysis, Investigation, Writing - original draft; Daisuke Motooka, Data curation, Formal analysis; Ian L Sparks, Methodology, Writing - original draft; Emily S Melzer, Data curation; Shota Nakamura, Resources, Supervision, Writing – review and editing; Enrique R Rojas, Resources, Supervision, Methodology, Writing – review and editing; Yasu S Morita, Conceptualization, Resources, Supervision, Funding acquisition, Validation, Investigation, Methodology, Project administration, Writing – review and editing; M Sloan Siegrist, Conceptualization, Resources, Data curation, Formal analysis, Supervision, Funding acquisition, Validation, Investigation, Visualization, Project administration, Writing – review and editing

## Author ORCIDs
Takehiro Kado (iD) http://orcid.org/0000-0001-5419-6123
Enrique R Rojas (iD) http://orcid.org/0000-0003-3388-2794
M Sloan Siegrist (iD) https://orcid.org/0000-0002-8232-3246

## Decision letter and Author response
Decision letter https://doi.org/10.7554/eLife.81924.sa1
Author response https://doi.org/10.7554/eLife.81924.sa2

## Additional files

### Supplementary files
- Supplementary file 1. Primers used in this study.
- Supplementary file 2. The Tn-seq result of Mann–Whitney *U*-test.
- MDAR checklist
- Source code 1. Code to analyze cell width profile.

### Data availability
Sequencing data have been deposited in NIH SRA under accession codes PRJNA976743. All data generated or analysed during this study are included in the manuscript and supporting file; Source Data files have been provided for all Figures and figure supplements.

The following dataset was generated:

| Author(s) | Year | Dataset title | Dataset URL | Database and Identifier |
|---|---|---|---|---|
| Kado T, Motooka D, Nakamura S, Morita YS, Siegrist MS | 2023 | Identification of over/under-represented genes after benzyl alcohol or dibucaine treatment in Mycobacterium smegmatis | https://www.ncbi.nlm.nih.gov/sra/PRJNA976743 | NCBI Sequence Read Archive, PRJNA976743 |

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
