## [Editor Report]

This paper addresses an important question: the relationship between the cell wall and other, primarily lipid, based components of the cell envelope. Building on previous work, the authors provide solid data suggesting that the activity of PonA2, a non-essential peptidoglycan synthase, promotes membrane partitioning through its role in cell wall synthesis. Altogether, this work provides valuable insight into the mechanisms coordinating the synthesis of separate layers of the bacterial cell envelope and as such should be of interest to microbiologists working on similar aspects of growth-related processes across bacterial systems.

---

## [Decision Letter]

**Decision letter after peer review:**

Thank you for submitting your article "The cell wall polymer initiates plasma membrane partitioning in mycobacteria" for consideration by *eLife*. Your article has been reviewed by 2 peer reviewers, and the evaluation has been overseen by a Reviewing Editor and Vivek Malhotra as the Senior Editor. The reviewers have opted to remain anonymous.

Essential revisions:

Both reviewers agreed that the idea that PG composition is a determinant of membrane organization, is potentially quite interesting. At the same time, they felt that the data in the current version of the manuscript is insufficient to make this point, particularly with regard to the role of PonA2. If PonA2 impacts membrane organization via its GTase activity, then glycan strand length should be perturbed in its absence. This needs to be tested directly (currently only data on crosslinking % is included). Additionally, the reviewers request experiments to resolve confusion about π staining and would like additional information about the role of DivIVA which is mentioned early in the manuscript but never resolved. Other experimental concerns are detailed in the body of the reviews below. Although not experimental in nature, both reviewers felt the term intracellular membrane domain (IMD) was confusing as it implies an intracellular phenomenon rather than different fluidities within the 2D space of the membrane.

*Reviewer #1 (Recommendations for the authors):*

In the discussion the authors clearly state (line 356-9) that it is the pre-existing wall, not active synthesis by PonA2, that is required for partitioning. However, at other places in the text, eg summary line 34-5, introduction line 112-113, header line 184, lines 229-230 (not exhaustive) the text can easily be interpreted (at least by me) as stating that PonA2 actively contributes to repartitioning during the repartitioning process. Of course, based on the genetic experiment this is a possibility, which later is investigated and found not to be the case, but more careful phrasing may help the reader understand the salient points better.

Figure 1 contains several images that are not critical to the important conclusions of the paper, but that do not (completely) support statements in the text:

– SDS and Oleic acid treatment result in an almost complete disappearance of fluorescent signal for Ppm1 and MurG (F1A). On the basis of these images, I don't think one can conclude that there is specific delocalization from the IMD under these conditions. It seems that mCherry does not function well when cells are exposed to these chemicals (especially since signal recovers after wash out – although this can also be from de novo synthesis).

– In the F1A right panel, GlfT2 does NOT seem delocalized from the IMD upon benzylalcohol treatment. It does seem delocalized in the left panel. Also Figure 1C seems to indicate that GlfT2 does not delocalize very strongly upon various treatments. Although beyond the scope of the paper, the extent of delocalization may be informative of how exclusively/strictly IMD associated some of these proteins are.

– The legend should state that F1C contains traces from cells imaged immediately after treatment to help the reader.

– Finally, what's the effect of DMSO on polarity of marker proteins? Figure 1 shows non-treated cells (top panels), but no DMSO control.

The authors should state the concentration of the DMSO vehicle control (2%? – 5M benzyl alcohol stock in DMSO; 100 mM final concentration).

L180: it would be helpful if the text explained how the ponA2 complementation was done – was ponA2 inserted at a specific ectopic site, and how is expression controlled? (the latter can be deducted from materials section but in my view it is better to state it in the main text).

Figure 4D – how is the material visualized? What are we looking at? There are only references to earlier published material whereas a short description of the method and visualization would be helpful for readers trying to understand what they are looking at. Also, and this may have to do with the image quality, the images provided do not sufficiently convince that there is a significant difference between the 'recovered' wild-type sample after 3 h wash out and the other treated samples.

Recovery after benzyl alcohol – why do the authors switch from measuring recovery via growth as measured via OD600 to measuring growth (Figure 3B) to growth as expressed by NADH production (resazurin assay, Figure 5A)? No explanation for the switch is given. It would be better if Figure 5A depicted a growth experiment similar to the one in Figure 3B, also because the DMSO treated samples show a reduction of growth for the two strains that are TG-.

Lines 711-715 – this describes a staining method that I did not find back in the paper – the method refers to the experiment in F4S2 where GFP expressing cells were used.

Figure 5B,C – The polarities of which IMD marker were determined? In the legends this is not mentioned, but for Figure 4 the text states that it's PPm1. For Figure 5 this is not clear (and if it's not PPm1 it should be clarified why a different marker was chosen).

The authors are completely correct in stating that π staining does not need to be linked to dead cells, there is quite some evidence for this in the literature including a study on a Mycobacterium species (https://doi.org/10.1002/cyto.a.20402). Adding this may help to make the point that the method is valid and shorten the section (details could still be provided in the legend to the supplemental figure).

*Reviewer #2 (Recommendations for the authors):*

An increase in membrane permeability, by a defect in PonA2 (and subsequent PG crosslinking) would still leave the question how the arabinogalactan and the mycolic acid layer contribute. Is the mycolic acid layer also compromised after benzyl alcohol treatment. How can we actually dissect the effect of the fluidizer on the two membranes (mycolic acid membrane and plasma membrane)?

I am a bit surprised about the data with propidium iodide. Based on the results shown here the dye seems not suited to monitor lethal membrane damage. The authors did some controls to show that their cells are still viable. Is it possible to monitor growth of π stained cells on agar pads or microfluidics?

Line 143: Loss of a specific protein localization and regaining this localization can have to do with membrane (re-)partitioning, but does not need to. At best this is indirect proof for membrane partitioning.

Line 153: six gene – six genes.

Line 466 (legend Figure 1): IMD fusions were imaged… The term IMD fusion is misleading. Fusion proteins were used and nothing fused to the membrane domains.

Table 1: The list of genes identified is quite diverse and it is difficult to see the common denominator. The antiporter could link to osmotic stress. Did the authors check any influence of these genes or osmotic stress?

Lien 182: The date show that PonA2 helps Msmeg to recover from benzyl alcohol treatment – I am sure that there is no membrane disruption (maybe a different membrane partitioning).

Figure 4A: Indicate in the figure that Ppm1-mNG is used. Also arrangement of Figure 4 is difficult to understand. 4C is divided in an upper and lower part, leaving the lower part next to 4D. The sucrose gradient data in 4D are difficult to decipher. What is the reader supposed to look at – the lighter areas, or the darker bands? The authors show data for 3 h after wash out of benzyl alcohol. The growth curves show that in this time the δ ponA2 mutants are still in lag phase. Thus, although the reason for the lag phase might be the problem in membrane compartmentalization, but it may also be an indirect effect – no elongation growth, no restoration of the membrane organization. We might look at a chicken and egg problem here that is difficult to solve.

Figure 4 supplement 2: What do the authors mean be "High-resolution microscopy"? Further, to show the plasmolysis a membrane stain would be better suited. A bump or lack of cytosolic GFP can have several reasons (septum formation, protein aggregation etc.).

Figure 5A: Why is here a Resazurin based growth curve and not OD as in Figure 3B. I find it difficult to use different techniques in a comparison. How would these growth curves look when OD600 is used? How did the authors check whether the point mutations have the phenotype they expect (no TG and no TP activity, respectively)?

Figure 7B. The legend states that for the untreated sample 67 cells were counted and for the treated ones 151 cells. The plot actually looks as if this is the other way round. I also find these data not robustly significant. If the same data set (and higher numbers) would be collected, it could very well be insignificant. This is important, because based on this finding the authors claim that cell membrane partitioning depends on the preexisting cell wall polymer.

Figure 8 is not well referenced in the text (only in line 435). The individual steps in the model are discussed in the text and therefore the authors can refer better to the model.

[Editors' note: further revisions were suggested prior to acceptance, as described below.]

Thank you for resubmitting your work entitled "A cell wall synthase accelerates plasma membrane partitioning in mycobacteria" for further consideration by *eLife*. Your revised article has been evaluated by Vivek Malhotra (Senior Editor) and a Reviewing Editor.

The manuscript has been improved but there are some remaining issues that need to be addressed, as outlined below:

Overall the reviewers were happy with the revisions and felt that this manuscript was significantly improved over the previous version. Although no additional experiments are necessary, some of the figure legends need to be revised for clarity (e.g. Figure 4, Figure 8 and a few others) and a few areas of the text would also benefit from minor revision. See reviewer comments below for details.

Reviewer #1 (Recommendations for the authors):

The authors have done a great job in revising the manuscript and addressing the issues raised by the reviewers. The relationship between PonA2 and membrane partitioning is novel, interesting and intriguing, and considerably strengthened by the additional work performed.

The authors have addressed all my questions. There is one thing that I would like to have seen worked out in more detail – but this is maybe also due to a lack of complete clarity on my part of what I would have liked to see in my original review.

This concerns the role of DivIVA – I asked for a recovery experiment with a DivIVA depletion strain. Although such an experiment is included, it only addresses recovery of growth (Figure 2 – Figure S1). I assume this experiment shows the recovery of cells after benzylalcohol wash out even though this is not clearly stated in the legend (or are these cells grown with benzylalcohol?). Nevertheless, what this experiment shows is that cells depleted of DivIVA recover growth more slowly than cells in which DivIVA is expressed. What I had hoped to see was an experiment similar to the ones presented in Figures4 and 6 where the role of DivIVA in the recovery of localization of a polar marker protein was studied. Such an experiment would strengthen (or refute) the author's notion that DivIVA helps maintain, but not establish, membrane partitioning.

---

## [Author Response]

Essential revisions:Both reviewers agreed that the idea that PG composition is a determinant of membrane organization, is potentially quite interesting. At the same time, they felt that the data in the current version of the manuscript is insufficient to make this point, particularly with regard to the role of PonA2. If PonA2 impacts membrane organization via its GTase activity, then glycan strand length should be perturbed in its absence. This needs to be tested directly (currently only data on crosslinking % is included).

Thank you for the feedback. Based in part on our new data (Figure 8B), we think there are three models for how PonA2 accelerates membrane partitioning (Figure 9). One of these is via cell wall integrity (Figure 9A). Within this model, however, we think that PonA2 could regulate glycan chain length, as suggested by the reviewer, and/or glycan order (PMID 29593214) and/or cell wall thickness. To add to the complexity, our new data (Figure 8B) suggest a role for active ongoing cell wall synthesis in PonA2-accelerated membrane partitioning. Thus, two additional, again not mutually exclusive, contributions of PonA2 may include consumption of lipid II, a molecule linked to membrane disorder in in vitro studies (Figure 9C), and/or polymerization of nascent strands of peptidoglycan, which transiently span the plasma membrane and cell wall, creating physical tethering (Figure 9A).

We are in the process of testing these hypotheses, including a new collaboration to quantitate glycan chain length in ∆*ponA2* and in other transglycosylase mutants that we have in hand (glycan chain length analysis has not yet been reported for mycobacteria). However, even if loss of PonA2 results in one or more of the phenotypes suggested by the foregoing hypotheses—shorter glycan chains, disordered glycan chains, thinner cell wall, lipid II accumulation, nascent peptidoglycan strand depletion—it will be a substantial undertaking to move beyond correlation and link these defects (or not) to membrane partitioning. To address, we will need to develop PonA2-independent ways to vary glycan chain length, glycan order, cell wall thickness, and lipid II and nascent peptidoglycan strand abundance. We think this is beyond the scope of the current manuscript.

We discuss our updated models (Figure 9) for how PonA2 accelerates membrane partitioning in the Discussion.

Additionally, the reviewers request experiments to resolve confusion about π staining and would like additional information about the role of DivIVA which is mentioned early in the manuscript but never resolved.

We have updated our analysis of π staining (Figure 7 —figure supplement 1) and performed an experiment to test the role of Wag31/DivIVA in membrane repartitioning postfluidization (Figure 2 —figure supplement 1). More details in our answers below.

Other experimental concerns are detailed in the body of the reviews below. Although not experimental in nature, both reviewers felt the term intracellular membrane domain (IMD) was confusing as it implies an intracellular phenomenon rather than different fluidities within the 2D space of the membrane.

The term “intracellular” is distinct from “intracytoplasmic”. When we named the membrane domain, we did not have sufficient evidence to indicate that the IMD is a laterally discrete domain of the plasma membrane. However, we agree with the reviewers that we now have accumulating evidence to claim the IMD as a domain within the 2D space of the plasma membrane. Therefore, we have changed the name to inner membrane domain (IMD). Please see line 70-71.

Reviewer #1 (Recommendations for the authors):In the discussion the authors clearly state (line 356-9) that it is the pre-existing wall, not active synthesis by PonA2, that is required for partitioning. However, at other places in the text, eg summary line 34-5, introduction line 112-113, header line 184, lines 229-230 (not exhaustive) the text can easily be interpreted (at least by me) as stating that PonA2 actively contributes to repartitioning during the repartitioning process. Of course, based on the genetic experiment this is a possibility, which later is investigated and found not to be the case, but more careful phrasing may help the reader understand the salient points better.

After discovering that addition of albumin changes our benzyl alcohol outgrowth phenotype for ∆*ponA2* (please see above and Materials and methods lines 735-743 for details) we repeated the D-cycloserine and moenomycin experiments in the absence of albumin and found that moenomycin (but not D-cycloserine) modestly dampens *M. smegmatis* recovery from benzyl alcohol. These new data (Figure 8B) and reinterpretation of old data (Figure 5C-D) support a more expansive view on the role(s) of cell wall synthesis/structure in membrane partitioning.

We have updated our model (Figure 9) and reorganized and revised the manuscript accordingly. Some examples:

1. Title: “A cell wall synthase accelerates plasma membrane partitioning in mycobacteria”.

2. Summary: “Active cell wall polymerization promotes de novo membrane partitioning and the completed cell wall polymer helps to maintain membrane partitioning.” lines 37-40

3. Significance: “We show that de novo membrane partitioning is accelerated by the transglycosylase domain of a bifunctional cell wall synthase.” lines 47-48

4. Introduction: “While the cell wall glycan backbone helps to maintain membrane partitioning, our data suggest that the role of PonA2 is specific to de novo partitioning and occurs at least in part via active cell wall polymerization.” lines 120-124

5. Results

New sub-heading “The cell wall glycan backbone contributes to membrane partitioning maintenance*.”* lines 364

New sub-heading *“*de novo membrane partitioning is supported by concurrent cell wall polymerization*.”* lines 378

“These data indicate that PonA2 is dispensable for maintaining membrane partitioning under unstressed conditions, at least to the limit of our detection. Therefore, we focused on understanding the role of PonA2 in de novo membrane partitioning.” lines 384-387

6. Discussion: “By dissecting the role of PonA2 in the presence or absence of benzyl alcohol-induced membrane disruption, we find that active cell wall polymerization helps to establish membrane partitioning, which is then maintained at least in part by the completed cell wall polymer.” lines 427-430

Figure 1 contains several images that are not critical to the important conclusions of the paper, but that do not (completely) support statements in the text:– SDS and Oleic acid treatment result in an almost complete disappearance of fluorescent signal for Ppm1 and MurG (F1A). On the basis of these images, I don't think one can conclude that there is specific delocalization from the IMD under these conditions. It seems that mCherry does not function well when cells are exposed to these chemicals (especially since signal recovers after wash out – although this can also be from de novo synthesis).

We agree and have removed the SDS and oleic acid data from Figure 1.

– In the F1A right panel, GlfT2 does NOT seem delocalized from the IMD upon benzylalcohol treatment. It does seem delocalized in the left panel. Also Figure 1C seems to indicate that GlfT2 does not delocalize very strongly upon various treatments.

We agree. Additionally, we note that (1) benzyl alcohol takes ~2 hours to delocalize GlfT2 (not shown); ~1 hr to delocalize Ppm1 (not shown); and 5 minutes to delocalize MurG (PMID: 33544079) and that (2) cell wall damage (1 hr treatment with lysozyme/mutanolysin) delocalizes Ppm1 (new Figure 8A) and MurG (PMID: 35543537) but not GlfT2 (new Figure 8-figure supplement 1). We still do not know how different proteins associate with different regions of the plasma membrane but these and other observations suggest that the nature and/or strength of these associations may differ. Additionally, or alternatively, some proteins may actively establish and/or maintain the membrane milieu, while others are recruited to the milieu once generated. We have included these ideas the text (lines 143-147 and 373-374) and included the new data in Figures 8A and Figure 8-figure supplement 1.

Although beyond the scope of the paper, the extent of delocalization may be informative of how exclusively/strictly IMD associated some of these proteins are.

We agree! In the future, we plan to do proteomics and lipidomics analyses of the IMD and PM-CW after treatment with benzyl alcohol or dibucaine, the latter a membrane fluidizer with a different proposed mechanism of action than benzyl alcochol (PMID: 33544079, 36976029). These experiments will help us to understand the association of different components of the plasma membrane under stress.

– The legend should state that F1C contains traces from cells imaged immediately after treatment to help the reader.

Fixed; please see line 518.

– Finally, what's the effect of DMSO on polarity of marker proteins? Figure 1 shows non-treated cells (top panels), but no DMSO control.

We have added this information as Figure 1-figure supplement 1.

The authors should state the concentration of the DMSO vehicle control (2%? – 5M benzyl alcohol stock in DMSO; 100 mM final concentration).

The concentration of DMSO vehicle control was 2% (v/v). See line 651.

L180: it would be helpful if the text explained how the ponA2 complementation was done – was ponA2 inserted at a specific ectopic site, and how is expression controlled? (the latter can be deducted from materials section but in my view it is better to state it in the main text).

Please see line 196-197.

Figure 4D – how is the material visualized? What are we looking at? There are only references to earlier published material whereas a short description of the method and visualization would be helpful for readers trying to understand what they are looking at. Also, and this may have to do with the image quality, the images provided do not sufficiently convince that there is a significant difference between the 'recovered' wild-type sample after 3 h wash out and the other treated samples.

These are images of the sucrose gradient fractions immediately after ultracentrifugation. The depletion and repletion of membranous material from the top (IMD) fractions +/- benzyl alcohol is striking by eye, but have been challenging to capture in a manuscript worthy format. We attempted to improve visualization by changing the order of images in Figure 4D (left) to make it easier to compare the tubes between wild-type and ∆*ponA2* and by performing densitometry analysis (right) similar to what we have done before (Figure 3 —figure supplement 1 in PMID: 33544079). We have updated the legend for Figure 4D and Materials and methods (line 763-772) accordingly.

Recovery after benzyl alcohol – why do the authors switch from measuring recovery via growth as measured via OD600 to measuring growth (Figure 3B) to growth as expressed by NADH production (resazurin assay, Figure 5A)? No explanation for the switch is given. It would be better if Figure 5A depicted a growth experiment similar to the one in Figure 3B, also because the DMSO treated samples show a reduction of growth for the two strains that are TG-.

We now include the following information in Materials and methods (lines 735-743):

“Our Tn-seq experiment (Figure 2) and initial validation (Figure 3B) were performed in 7H9 growth medium that lacked albumin. We found that *M. smegmatis* clumped when grown in the same medium in 96-well plate format, precluding accurate OD_600_ readings. Addition of albumin prevented clumping. However, genetic or chemical perturbation of PonA2 no longer had a phenotype post-benzyl alcohol when albumin was included in the growth medium. While we do not yet understand this effect, we speculate that albumin might mitigate benzyl alcohol-induced fluidization and/or alter osmolarity. Therefore, we used resazurin reduction as an alternative, high-throughput method to assess *M. smegmatis* growth +/- benzyl alcohol for the experiments in Figure 6A and Figure 8B.”

Lines 711-715 – this describes a staining method that I did not find back in the paper – the method refers to the experiment in F4S2 where GFP expressing cells were used.

We added the following to the Materials and methods line 817-821:

“In the case where cells were stained with RADA, a TAMRA-based fluorescent D-amino acid (Tocris Bioscience), 1 µM RADA was added to the culture upon back dilution. If the cells were not stained with RADA, Alexa Fluor 647 NHS succinimidyl ester (Thermo Fisher Scientific) was added to the media as an occlusion dye (it is not cell wall permeable and thus can be used to track the cells).”

Figure 5B,C – The polarities of which IMD marker were determined? In the legends this is not mentioned, but for Figure 4 the text states that it's PPm1. For Figure 5 this is not clear (and if it's not PPm1 it should be clarified why a different marker was chosen).

For Figure 4, Figure 5 (new Figure 6B-C), and new Figure 8A we used Ppm1 as our IMD marker. We have updated our legends.

The authors are completely correct in stating that π staining does not need to be linked to dead cells, there is quite some evidence for this in the literature including a study on a Mycobacterium species (https://doi.org/10.1002/cyto.a.20402). Adding this may help to make the point that the method is valid and shorten the section (details could still be provided in the legend to the supplemental figure).

We have added the reference (PMID: 17421025), thank you for pointing that out. We opted to keep the text in light of Reviewer 1’s concerns.

Reviewer #2 (Recommendations for the authors):An increase in membrane permeability, by a defect in PonA2 (and subsequent PG crosslinking) would still leave the question how the arabinogalactan and the mycolic acid layer contribute. Is the mycolic acid layer also compromised after benzyl alcohol treatment. How can we actually dissect the effect of the fluidizer on the two membranes (mycolic acid membrane and plasma membrane)?

Dissection of the inner plasma membrane independent of the outer (myco)membrane remains an outstanding challenge for both mycobacteria and Gram-negatives.

While we cannot rule out an effect on other parts of the cell envelope, we previously showed that pre-existing mycomembrane components are not delocalized upon benzyl alcohol treatment (Figure 3—figure supplement 4 in PMID: 33544079). In this experiment, wild-type *M. smegmatis* were incubated with O-AlkTMM (primarily labels covalent mycolates) or N-AlkTMM (labels noncovalent mycolates) then washed, and treated benzyl alcohol for 1 hour. The lack of change to the mycomembrane +/- benzyl alcohol was in contrast to the many effects that we observed for the plasma membrane.

However, we note that our preliminary data does suggest that mycomembrane synthesis delocalizes upon benzyl alcohol treatment (not shown), as we have demonstrated for peptidoglycan synthesis (Figure 3B in PMID: 33544079). This could be for several reasons, not mutually exclusive: benzyl alcohol directly disrupts the mycomembrane; benzyl alcohol disrupts arabinogalactan/mycomembrane precursor synthesis and/or localization within the plasma membrane; benzyl alcohol-induced disruption and delocalization of peptidoglycan synthesis mis-templates sites of new arabinogalactan/mycomembrane synthesis.

For these and other reasons, we are mindful that phenotypes that we see +/- benzyl alcohol may be pleiotropic. In the current study, we see clear effects of benzyl alcohol on IMD protein localization and IMD biochemical isolability. We think that a direct effect(s) of cell wall synthesis/structure on membrane repartitioning is the most parsimonious explanation for our PonA2 data (Figure 9) especially given the physical proximity of the cell wall and plasma membrane. However, it is certainly possible that altered cell wall synthesis/structure impacts the mycomembrane, which in turn feeds back on the plasma membrane. We have now included this idea in the Discussion, lines 442-445 and lines 464-465.

I am a bit surprised about the data with propidium iodide. Based on the results shown here the dye seems not suited to monitor lethal membrane damage. The authors did some controls to show that their cells are still viable. Is it possible to monitor growth of π stained cells on agar pads or microfluidics?

We were also surprised! However, we found reports in the literature suggesting that propidium iodide positivity was not always lethal. Per Reviewer 1’s suggestion, we have now included one of them, a study conducted with a mycobacterial species, in the manuscript (PMID: 17421025, line 338).

Taken as a whole, we think that the literature precedent, flow cytometry data in Figure 7—figure supplement 1B, CFUs in Figure 3A and Figure 3–figure supplement 1, and growth kinetics in Figure 3B together are consistent with our conclusion that PonA2 contributes to growth recovery from benzyl alcohol rather than resistance to lethal membrane damage.

Line 143: Loss of a specific protein localization and regaining this localization can have to do with membrane (re-)partitioning, but does not need to. At best this is indirect proof for membrane partitioning.

We changed the way we presented these data to clearly distinguish IMD protein localization (Figure 1A-C) vs. biochemical isolability (Figure 1D), both of which are impacted by benzyl alcohol (lines 143-147).

Line 153: six gene – six genes.

We fixed this.

Line 466 (legend Figure 1): IMD fusions were imaged… The term IMD fusion is misleading. Fusion proteins were used and nothing fused to the membrane domains.

We rephrased “IMD fusions” to “IMD proteins”.

Table 1: The list of genes identified is quite diverse and it is difficult to see the common denominator. The antiporter could link to osmotic stress. Did the authors check any influence of these genes or osmotic stress?

Our preliminary data suggest that osmotic shock (as in Figure 5) does not delocalize the IMD marker GlfT2. However, given that MurG and Ppm1 localization are consistently more sensitive to perturbation than GlfT2 localization (Figure 1D, Figure 8A, Figure 8—figure supplement 1, PMID: 35543537, and described in this document), we are hesitant to draw conclusions without further experimentation.

We have now confirmed 3/3 hits from this and a similar Tn-seq screen that we performed in parallel (using a different, membrane-disrupting chemical; PMID: 36976029 and not shown). However, we agree that it is a diverse list. We suspect that other genes that contribute to membrane (re)partitioning are essential, e.g., PonA1, or redundant, e.g., there are multiple flotillin homologs.

Line 182: The date show that PonA2 helps Msmeg to recover from benzyl alcohol treatment – I am sure that there is no membrane disruption (maybe a different membrane partitioning).

Now line 203, changed as suggested.

Figure 4A: Indicate in the figure that Ppm1-mNG is used. Also arrangement of Figure 4 is difficult to understand. 4C is divided in an upper and lower part, leaving the lower part next to 4D. The sucrose gradient data in 4D are difficult to decipher. What is the reader supposed to look at – the lighter areas, or the darker bands? The authors show data for 3 h after wash out of benzyl alcohol. The growth curves show that in this time the δ ponA2 mutants are still in lag phase. Thus, although the reason for the lag phase might be the problem in membrane compartmentalization, but it may also be an indirect effect – no elongation growth, no restoration of the membrane organization. We might look at a chicken and egg problem here that is difficult to solve.

We rearranged the panels, rephrased the legend for Figure 4D, and did densitometry analysis on the membranous material highlighted by the solid and dashed gray lines next to the tubes. We have also updated Materials and methods line 763-772.

We agree with the reviewer that the correlation between membrane partitioning and growth does not equal causation. We now highlight this important point in the Discussion: “Membrane partitioning correlates with active *M. smegmatis* growth, although a causative role of partitioning in growth remains speculative.” line 428-429.

Figure 4 supplement 2: What do the authors mean be "High-resolution microscopy"? Further, to show the plasmolysis a membrane stain would be better suited. A bump or lack of cytosolic GFP can have several reasons (septum formation, protein aggregation etc.).

This is now Figure 5. We have removed the term “high-resolution” and included before and after images to better showcase the bays.

Figure 5A: Why is here a Resazurin based growth curve and not OD as in Figure 3B. I find it difficult to use different techniques in a comparison. How would these growth curves look when OD600 is used?

As we discuss above, we now include the following information in Materials and methods (lines 735-743):

“Our Tn-seq experiment (Figure 2) and initial validation (Figure 3B) were performed in 7H9 growth medium that lacked albumin. We found that *M. smegmatis* clumped when grown in the same medium in 96-well plate format, precluding accurate OD_600_ readings. Addition of albumin prevented clumping. However, genetic or chemical perturbation of PonA2 no longer had a phenotype post-benzyl alcohol when albumin was included in the growth medium. While we do not yet understand this effect, we speculate that albumin might mitigate benzyl alcohol-induced fluidization and/or alter osmolarity. Therefore, we used resazurin reduction as an alternative, high-throughput method to assess *M. smegmatis* growth +/- benzyl alcohol for the experiments in Figure 6A and Figure 8B.”

How did the authors check whether the point mutations have the phenotype they expect (no TG and no TP activity, respectively)?

We used moenomycin sensitivity as a proxy for TG activity and assessed TP activity by Bocillin-FL. We have now included these data as Figure 6–figure supplement 2 and updated our Results line 280-283 Materials and methods lines 774-779 accordingly.

Figure 7B. The legend states that for the untreated sample 67 cells were counted and for the treated ones 151 cells. The plot actually looks as if this is the other way round. I also find these data not robustly significant. If the same data set (and higher numbers) would be collected, it could very well be insignificant. This is important, because based on this finding the authors claim that cell membrane partitioning depends on the preexisting cell wall polymer.

We thank the reviewer for their candor. We repeated the experiment to include additional biological and technical replicates in our analysis. As it turns out, the difference was not statistically significant (now Figure 8–figure supplement 1).

In retrospect, this perhaps should not have been a surprise given that the IMD marker that we chose to assess in this experiment (GlfT2) behaves differently from the other IMD markers (MurG and Ppm1) +/- benzyl alcohol (Figure 1C and discussed earlier in this document). We repeated the experiment with Ppm1, the IMD marker protein that we used as a proxy for IMD localization in other parts of the paper (Figures 4A-C and 6B-C), and found that there was indeed a modest decrease in polarity upon lysozyme/mutanolysin treatment (Figure 8A).

The decrease in Ppm1 polarity in response to cell wall damage is not as striking as what we previously observed for MurG (see Figure 3 in PMID: 35543537). It is possible that this reflects differences in marker protein association with the IMD, marker protein function, and/or marker protein expression (*e.g*., MurG-Dendra2, which is not expressed under its native promoter, is much brighter than Ppm1-mNeonGreen, which is expressed under its native promoter).

Figure 8 is not well referenced in the text (only in line 435). The individual steps in the model are discussed in the text and therefore the authors can refer better to the model.

We have updated our model (now Figure 9) to encompass our new data (Figure 8B) and refer to it the Discussion lines 433-445.

[Editors' note: further revisions were suggested prior to acceptance, as described below.]

The manuscript has been improved but there are some remaining issues that need to be addressed, as outlined below:Overall the reviewers were happy with the revisions and felt that this manuscript was significantly improved over the previous version. Although no additional experiments are necessary, some of the figure legends need to be revised for clarity (e.g. Figure 4, Figure 8 and a few others) and a few areas of the text would also benefit from minor revision. See reviewer comments below for details.Reviewer #1 (Recommendations for the authors):The authors have done a great job in revising the manuscript and addressing the issues raised by the reviewers. The relationship between PonA2 and membrane partitioning is novel, interesting and intriguing, and considerably strengthened by the additional work performed.The authors have addressed all my questions. There is one thing that I would like to have seen worked out in more detail – but this is maybe also due to a lack of complete clarity on my part of what I would have liked to see in my original review.This concerns the role of DivIVA – I asked for a recovery experiment with a DivIVA depletion strain. Although such an experiment is included, it only addresses recovery of growth (Figure 2 – Figure S1). I assume this experiment shows the recovery of cells after benzylalcohol wash out even though this is not clearly stated in the legend (or are these cells grown with benzylalcohol?). Nevertheless, what this experiment shows is that cells depleted of DivIVA recover growth more slowly than cells in which DivIVA is expressed. What I had hoped to see was an experiment similar to the ones presented in Figures4 and 6 where the role of DivIVA in the recovery of localization of a polar marker protein was studied. Such an experiment would strengthen (or refute) the author's notion that DivIVA helps maintain, but not establish, membrane partitioning.

Yes, the DivIVA experiment in Figure 2 —figure supplement 1 shows wild-type (left) or DivIVA mutant (right) grown after benzyl alcohol washout. The conditions were similar to those in Figures 2 and 3B. We have updated the legend in red to clarify.

And yes, we apologize, we misunderstood the original ask RE IMD marker protein experiment. We previously showed that DivIVA depletion correlates with delocalization of the IMD-enriched protein GlfT2 (Figure 4B in PMID: 33544079). More recently we’ve seen that this is reversible, i.e., DivIVA repletion corresponds with GlfT2 relocalization (unpublished). It’s a great idea to do the repletion after benzyl alcohol exposure as this is more specific to membrane partitioning and may be easier to interpret than the growth assay. We can do this experiment with the strain that we have in hand (DivIVA depletion strain that expresses GlfT2 fusion, as referenced above) and, given that the IMD markers don’t always behave the same way (e.g., Figure 1C, Figure 8A, and Figure 8 —figure supplement 1), we can also construct DivIVA depletion strains that express the Ppm1 or MurG fusions for this purpose.